



# From Sea to Sky: Understanding the sea surface temperature impact on an atmospheric blocking event using sensitivity experiments with the ICOsahedral Nonhydrostatic (ICON) model.

Svenja Christ[1], Marta Wenta[1], Christian M. Grams[1,2], and Annika Oertel[1]

[1]Institute of Meteorology and Climate Research, Troposphere Research (IMKTRO), Karlsruhe Institute of Technology (KIT), Karlsruhe, Germany
[2]now at Federal Office of Meteorology and Climatology, MeteoSwiss, Zurich, Switzerland

**Correspondence:** Svenja Christ (svenja.christ@kit.edu)

**Abstract.** Blocked weather regimes are an important phenomenon in the Euro-Atlantic region and are frequently linked to extreme weather events. Despite their importance for surface weather, the correct prediction of blocking events remains challenging. Previous studies indicated a link between the misrepresentation of blocking events in numerical weather prediction models and sea surface temperature (SST) biases, particularly in the Gulf Stream region. However, the pathway that links SST

in the Gulf Stream region and the downstream upper-level flow is not yet fully understood. To deepen our physical understanding of the link between the Gulf Stream SST and downstream atmospheric blocking, we perform sensitivity experiments with varying SST conditions for an atmospheric blocking event in February 2019. This blocking event, which was associated with a winter heatwave with unprecedented temperatures in Western Europe, was both preceded and accompanied by several rapidly intensifying extratropical cyclones originating in the Gulf Stream region and crossing the North Atlantic. Those cyclones and

their associated rapidly ascending air streams, so-called warm conveyor belts (WCBs), played a crucial role in the development of the upper-level ridge and the blocking event. The ascent of these WCBs, which connect the lower and upper troposphere, was enhanced by moisture uptake during cold air outbreaks (CAOs) in the Gulf Stream region. In this study, we employ sensitivity experiments with the Icosahedral Nonhydrostatic Weather and Climate Model (ICON) to assess the impact of intense air-sea interactions during CAOs on WCBs and the downstream ridge. In total five different experiments are used which in-

clude idealized and weakened SST gradients, and one with increased absolute SST in the Gulf Stream region. Using Eulerian and Lagrangian perspectives, we demonstrate that the SST gradient in the Gulf Stream region affects moisture availability and air temperature in the WCB inflow region, and consequently WCB ascent. In our case study, stronger SST gradients lead to increased specific humidity and warmer temperatures in the lower troposphere, resulting in more pronounced WCB ascent, while weaker SST gradients are associated with reduced WCB activity. The differences in WCB ascent and outflow properties

induced by weakened SST gradients, such as reduced cross-isentropic ascent and outflow heights, subsequently influence the upper-level flow and weaken the downstream ridge. Moreover, experiments with weaker SST gradients show a decrease in cyclone intensity, and vice versa, stronger cyclones are found in experiments with warmer SST. To summarize, our results suggest that different SST and SST gradient representations affect the large-scale atmospheric flow via the WCB airstream. Specifically, moisture availability regulated by SST and SST gradients in the WCB inflow region influences subsequent WCB





ascent and outflow characteristics which, in turn, influences the upper-level ridge downstream. The SST in the Gulf Stream region affects WCB characteristics consistently from the inflow, over the ascent to the outflow phase.

# 1 Introduction

Blocking regimes form a persistent, quasi-stationary atmospheric state that interrupts the eastward propagation of mid-latitude weather systems (Michelangeli et al., 1995; Teubler and Riemer, 2016). They are often linked to extreme weather events (e.g.,

Yiou and Nogaj, 2004; Booth et al., 2017; Schaller et al., 2018; Spensberger et al., 2020; Kautz et al., 2022), as the associated persistent anticyclonic circulation can dominate the weather at a particular location for several days to weeks (Wazneh et al., 2021). To date, the correct prediction of blocking events and associated surface weather remains challenging (Grams et al., 2018; Büeler et al., 2021; Oertel et al., 2023b). Previous research linked the Gulf Stream sea surface temperature (SST) with downstream blocking events in the North Atlantic and Western Europe (e.g. O'Reilly et al., 2016; Yamamoto et al., 2021)

and suggested that SST biases in the North Atlantic region might contribute to challenges in predicting blocking downstream (Czaja et al., 2019; Roberts et al., 2021; Athanasiadis et al., 2022). However, the physical pathway between SST in the Gulf Stream region and atmospheric blocking events is still not fully understood. In their seminal work Pfahl et al. (2015) showed that latent heat release in ascending air streams is a first-order process in the development of blocking anticyclones. Most of this ascent is confined to the warm conveyor belt (WCB) airstream associated with extratropical cyclones (e.g. Madonna et al.,

2014) which ascends into the upper troposphere where it can amplify upper-level ridges subsequently resulting in blocking (e.g. Grams and Archambault, 2016). WCB airstreams occur frequently in the mid-latitudes and accompany approximately 60% of extratropical cyclones (Carlson, 1980; Eckhardt et al., 2004). The Western North Atlantic, particularly the Gulf Stream region, is one of the hotspots for WCB development (Eckhardt et al., 2004; Madonna et al., 2014). WCBs originate in the lower troposphere in the warm sector of extratropical cyclones and typically ascend poleward near the surface cold front

(Wernli, 1997). SST variability can affect air-sea interactions and thus modulate low-level moisture in the WCB inflow region, which subsequently affects the associated WCB ascent and outflow characteristics in the upper troposphere (e.g. Schäfler and Harnisch, 2015; Yamamoto et al., 2021; Wenta et al., 2024). In this study, we thus investigate the hypothesis that the WCB airstream links SST variations in the Gulf Stream region to downstream blocking events. In the following, we provide a brief review of previous studies investigating the influence of SST on cyclone dynamics, cold air outbreaks (CAO), and the impact

of WCBs on blocking events. Furthermore, we summarize the impact of moisture availability on WCBs and diabatic heating.

Extratropical cyclones in the North Atlantic and Pacific Oceans tend to organize along strong SST gradients (Nakamura et al., 2004). In the North Atlantic, the largest frequency of mid-latitude atmospheric fronts occurs along the Gulf Stream SST gradient (Berry et al., 2011; Reeder et al., 2021). As a consequence, the influence of SST changes on cyclones is most pronounced in these regions (e.g., Small et al., 2014; Tsopouridis et al., 2021). For example, cyclone intensification is sensitive

to latent heat fluxes in their vicinity, which is directly influenced by SST (Vries et al., 2019). Modifying the SST gradient in global atmospheric models has been shown to impact not only the location but also the characteristics of the storm track (Small et al., 2014). The maintenance of the storm track itself is linked to increased low-level baroclinicity (Hotta and Nakamura,



2011; Papritz and Spengler, 2015), which can be sustained in regions like the Gulf Stream due to differential surface heat fluxes (Papritz et al., 2015). On subseasonal-to-seasonal timescales, forecasts could benefit from a higher ocean resolution

with improved representation of small ocean eddies (Roberts et al., 2022), while the small-scale SST perturbations may have a relatively small influence on individual synoptic events (Roberts et al., 2021). Nevertheless, Tsopouridis et al. (2021) reported reduced cyclone activity in the North Atlantic as a result of a smoothed SST gradient. In contrast, Bui and Spengler (2021) observed no notable differences in the deepening rates of individual cyclones between smoothed and observed SST conditions. Instead, Bui and Spengler (2021) concluded that the distribution of absolute SST, rather than the SST gradient, influences latent

heat release and subsequent diabatic cyclone intensification. In line with this, numerical experiments suggest that cyclones respond to a decreasing SST gradient differently, depending on whether the cyclones pass over the warm or cold site of the SST gradient (Booth et al., 2012). Overall, this emphasizes the importance of the Gulf Stream for cyclone dynamics in the North Atlantic but also suggests that the influence of absolute SST and SST gradients on the synoptic flow evolution is not yet fully understood.

The passage of cyclones across the Gulf Stream is often linked to the development of CAOs and intense air-sea interactions, driven by the large air-sea temperature differences (e.g., Papritz and Spengler, 2015). The resulting surface heat and moisture fluxes play an important role in regulating the heat and moisture supply to the rapidly ascending air streams (WCBs; Booth et al., 2017). Warmer SSTs enforce more latent heat fluxes and potentially enhance latent heat release and cyclone intensification (Booth et al., 2012; Bui and Spengler, 2021), whereby more intense cyclones tend to have stronger WCBs (Binder et al.,

2016). Due to its deep ascent, the WCB airstream connects the boundary layer and the upper troposphere (Wernli, 1997). The ascent of the WCB is characterized by substantial latent heat release (Browning, 1990) and cloud bands forming during WCB ascent can reach lengths of up to 3000 km (Browning et al., 1973). Latent heating enables the cross-isentropic flow of WCB air parcels (Wernli and Davies, 1997; Joos and Wernli, 2012; Madonna et al., 2014). Moreover, the latent heating pattern along the ascent is linked to the characteristic potential vorticity (PV) changes along WCB trajectories, as PV is produced below the

level of maximum latent heat release and reduced above (Wernli and Davies, 1997; Madonna et al., 2014). As a consequence, the WCB performs a net transport of low PV air into the upper troposphere (Wernli and Davies, 1997), where it can contribute to the formation of a negative PV anomaly (Wernli and Davies, 1997; Grams et al., 2011; Joos and Forbes, 2016). Additionally, the divergent outflow of the WCB contributes to the anticyclonic circulation in the upper troposphere (Browning and Roberts, 1994; Steinfeld and Pfahl, 2019). For these reasons, the divergent WCB outflow can play an important role in the mainte-

nance and intensification of the downstream blocking event (Grams et al., 2011; Michel and Rivière, 2011; Pfahl et al., 2015; Grams and Archambault, 2016; Teubler and Riemer, 2016; Steinfeld and Pfahl, 2019; Steinfeld et al., 2020). As WCB ascent substantially influences ridge amplification and blocking intensity, the detailed representation of WCB ascent is important to correctly represent flow properties and minimize forecast error (e.g., Schäfler and Harnisch, 2015; Grams et al., 2018; Berman and Torn, 2019; Steinfeld et al., 2020; Berman and Torn, 2022; Pickl et al., 2023). A key factor for the strengthening of WCB

ascent is moisture availability since this can influence subsequent latent heat release and cross-isentropic ascent (Schäfler and Harnisch, 2015; Dacre et al., 2019; Quinting and Grams, 2021; Berman and Torn, 2022; Quinting et al., 2022; Oertel et al., 2023a). Thus, stronger latent heating in response to climate change is expected to influence the size and intensity of blocking





anticyclones (Steinfeld et al., 2022). Further, the releases of latent heat can introduce forecast uncertainties (Berman and Torn, 2019, 2022). Therefore, the intensity of the WCB and its impact on PV values in the upper troposphere are closely linked to
moisture availability in the lower troposphere (Schäfler and Harnisch, 2015; Schemm et al., 2013).

A primary moisture source for extratropical cyclones is local and located over the Western North Atlantic, specifically in the Gulf Stream region (Pfahl et al., 2014; Papritz et al., 2021). In this region, initial cold and dry air from the American continent is heated and moistened by the warm waters south of the SST front during CAOs (Papritz et al., 2021). The advection of cold air across the Gulf Stream leads to intense surface heat fluxes. The moistening of the marine boundary layer is often caused by the
passage of a predecessor cyclone, pointing at a possible cyclone-to-cyclone interaction (Sodemann et al., 2008). Specifically, Dacre et al. (2019) and Papritz et al. (2021) demonstrated that the cyclone's moisture typically originates in the cold sector in the pre-cyclone environment of the preceding cyclone. In particular, CAOs caused by the preceding cyclone contribute to this moisture uptake (Wenta et al., 2024).

The described synoptic features are relevant for the synoptic evolution investigated here. In February 2019, an atmospheric
blocking event led to the development of a significant winter heatwave in Western Europe (Kendon et al., 2020; Young and Galvin, 2020; Leach et al., 2021). This event was both preceded and accompanied by a series of rapidly intensifying cyclones, leading to the occurrence of intense CAOs in the Western and Central North Atlantic (Wenta et al., 2024). Wenta et al. (2024) indicated that these cyclones and their associated WCBs may have contributed to the development and maintenance of the block, as they played a key role in forming the upper-level PV anomaly associated with the European blocking event. Further-
more, Wenta et al. (2024) showed that the moisture sources for those cyclones were associated with air-sea interactions during CAOs initiated by the preceding cyclones in the same region, in agreement with the conceptual model proposed by Papritz et al. (2021). These results suggest that cyclones and their associated WCB airstreams provide a mechanistic link of surface processes in the Gulf Stream region with the development of an atmospheric block downstream in February 2019 (see also Kwon et al., 2010; Czaja et al., 2019; Athanasiadis et al., 2022; Wenta et al., 2024).

In this study, we want to explore how sensitive this mechanistic link is to variations of absolute SST and SST gradients in the Gulf Stream region, specifically how absolute SST and SST gradients modulate air-sea interaction, subsequent cyclone development, WCB ascent, and upper-level flow amplification. Therefore we conduct numerical sensitivity experiments using the ICON (ICOsahedral Nonhydrostatic) model and address the following main research questions:

1. How do SST gradient and absolute SST perturbations in the Gulf Stream region influence air-sea interactions?

2. How do changes in low-level moisture availability from SST perturbations influence the ascending WCB airstream linking the lower and upper troposphere?

3. Do air-sea interactions over the Gulf Stream region influence upper-level ridge amplification and the formation of the European blocking event through their impact on WCB ascent associated with extratropical cyclones?

In the following section, we provide a detailed description of the methodology (Section 2). This is followed by a more
detailed introduction to the February 2019 case study (Section 3) and the presentation of the results (Section 4). We finally conclude with a discussion and summary of the results (Sektion 5).





## 2   Methods

To examine the impact of SST in the Gulf Stream region on WCB ascent and the subsequent development of a downstream ridge, we conduct a detailed case study of a European blocking event (Section 3 for details). The blocking event in February 2019 was preceded and accompanied by a series of rapidly intensifying cyclones, each associated with a WCB. To determine how variations in SST and SST gradient in the Gulf Stream region influence WCB characteristics, numerical simulations with the ICON model (ICOsahedral Nonhydrostatic Weather and Climate Model, Zängl et al., 2015) were performed, featuring five different prescribed SST configurations in the Eastern North Atlantic (see Section 2.2 for details).

### 2.1   ICON model setup

The ICON simulations (version 2.6.2.2) are run freely and globally for nine days lead time. The simulations are run with a horizontal resolution of about 13 km (R3B07 grid), along with 90 vertical model levels and a time step of 120 s. This setup corresponds to the operational resolution of the global ICON model used by the German weather service. The ICON simulations are initialized from ECMWF's IFS analysis at 00 UTC 18 February 2019. Cloud microphysical processes in the model are represented using a single-moment scheme (Seifert, 2008; Doms et al., 2018), which includes four prognostic hydrometeor categories: cloud water, rain, ice, and snow. The model employs a Tiedtke–Bechtold bulk mass flux scheme for parametrizing convection (Tiedtke, 1989; Bechtold et al., 2008), and the radiation calculations are performed on a reduced radiation grid (R3B06 grid) utilizing the ecRAD scheme (Hogan and Bozzo, 2018). Additionally, the default ICON schemes are applied for sub-grid-scale orographic drag (Lott and Miller, 1997), non-orographic gravity wave drag (Orr et al., 2010), and turbulence (Raschendorfer, 2018). Surface fluxes are parameterized using the drag-law formulation (Raschendorfer, 2018), which takes into account the horizontal velocity at the lowest model level, the bulk-aerodynamical transfer coefficient for turbulent heat exchange at the surface, and the surface temperature.

### 2.2   Design of sensitivity experiments

In total, five separate sensitivity experiments with modified SST are conducted, to address the research questions. The reference SST is derived from the ECMWF IFS analysis and represents large SST gradients associated with the Gulf Stream (Fig. 1a). Throughout each of the nine-day long simulations, the prescribed SSTs remain constant. The five experiments are characterized by the following modifications to the reference SST pattern:

1. *Control experiment (CNTRL)*
   The simulation is initialized with SST from the ECMWF IFS analysis (Fig. 1a) which has been remapped to the ICON grid. Due to the relatively high resolution of around 9 km smaller-scale meanders and locally large SST gradients are present.

2. *Idealized SST gradient experiment (IDEA)*
   For IDEA, the SST front's small-scale eddies are removed (Fig. 1b) to create an idealized and smooth SST gradient. To





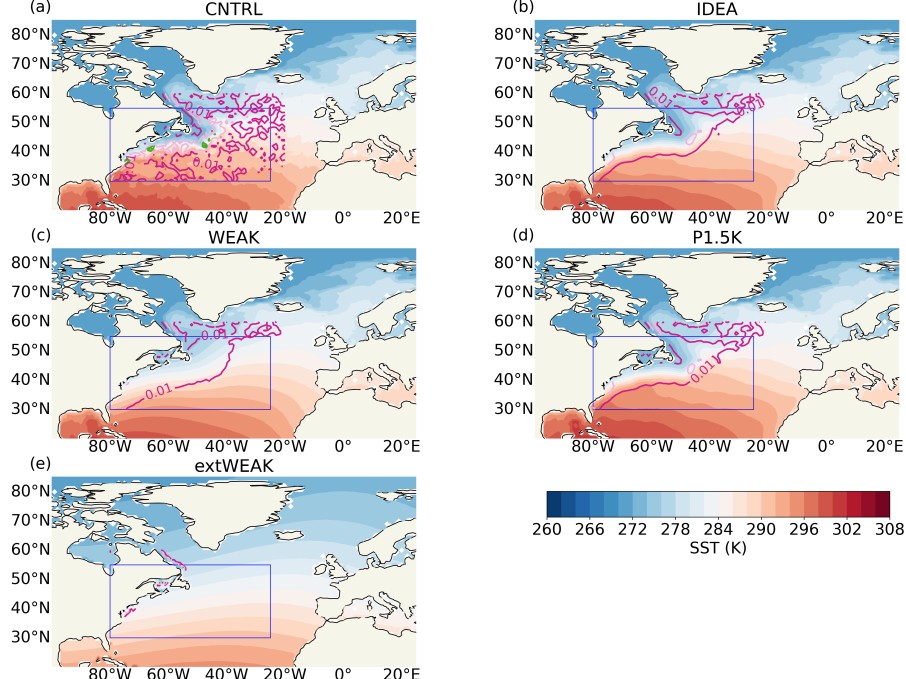

**Figure 1.** Sea surface temperature (SST, shading, in K) and SST gradient (contours, at 0.01, 0.05, and 0.1 K km$^{-1}$ colored in magenta shades) for the nine-day simulation period for the experiments **(a)** CNTRL, **(b)** IDEA, **(c)** WEAK, **(d)** extWEAK, and **(e)** P1.5K. The Gulf Stream region referred to in the text is outlined by the blue box (30 to 55 ° N and 80 to 25 ° W).

achieve this, we apply a Gaussian filter to the two-dimensional SST field in the Gulf Stream region (30 to 55° N and 80 to 25° W). This filter uses a Gaussian kernel with a standard deviation of three, applied uniformly in both directions. To prevent artificially strong temperature gradients at the border regions of the modified SST field, we included additional smoothing at the border region. As IDEA exhibits only minor differences compared to CNTRL, providing nearly identical results (see Section 4), this experiment serves as a reference throughout the study.

3. *Weak SST gradient experiment (WEAK)*

For WEAK (Fig. 1c), a stronger smoothing than for IDEA was applied with a standard deviation of 12 for the Gaussian kernel applied in the Gulf Stream region. This results in a substantially weakened SST gradient at the Gulf Stream SST front.

4. *Extra weak SST gradient experiment (extWEAK)*

In the extWEAK simulation, an even stronger smoothing with a standard deviation of 36 for the Gaussian kernel essentially removed the distinct SST front (Fig. 1e). Moreover, SST was altered in a broader area in the North Atlantic to prevent artificial high SST gradients at border regions. In this setup, the SST transitions gradually and cools progressively from the equator towards the higher latitudes.



**Table 1.** Averaged SST (mean and standard deviation) in the Gulf Stream region (30 to 55 ° N and 80 to 25 ° W, see blue box in Fig. 1) for the five different experiments.

|  | SST | Characteristics |
|---|---|---|
| CNTRL | 286.38±7.42 K | SST is taken from the IFS analysis |
| IDEA | 286.34±7.04 K | small-scale meanders are removed; local SST differences prevail |
| WEAK | 285.98±5.64 K | SST gradient is smoothed; cooler SST south of the SST front and warmer SST north of the SST front occur |
| extWEAK | 285.12±3.57 K | strong smoothing removes the SST front; substantially cooler SST south of the SST front and warmer SST north of the SST front prevail |
| P1.5K | 287.27±6.99 K | SST is increased by up to 1.5 K around the Gulf Stream front |

5. *Warmed SST experiment (P1.5K)*

Besides the experiments involving modified SST gradients, the P1.5K experiment features an increase in SST by up to 1.5 K (Fig. 1d). Specifically, SST is increased by 1.5 K in an ellipse centered around the Gulf Stream SST gradient at 41.5 ° N and 62.5 ° W. The applied warming gradually decreases towards domain boundaries to prevent the generation of artificial gradients elsewhere in the North Atlantic. In effect, the applied warming is limited to 35 to 55 ° N and 40 to 80 ° W.

The SST modifications slightly change the area-mean SST in the region of interest located between 30 to 55 ° N and 80 to 25 ° W (Tab. 1). This region is of particular interest as most of the cyclones associated with the block pass through, and often intensify, in this region. As expected, CNTRL and IDEA show almost no discernible difference in area-averaged SST, with the only alteration between those experiments being the removal of the small-scale meanders in the Gulf Stream. Per definition, P1.5K has the warmest area-averaged SST of 287.27 K (Tab. 1) and higher SST in the entire Gulf Stream region with maximum differences reaching up to 1.5 K compared to IDEA. In the WEAK and extWEAK simulations, the SSTs are warmer north of the SST front and colder to the south in comparison to IDEA. As the area south of the SST front in the region of interest is larger, the average SST in the Gulf Stream region is effectively lower in both the WEAK and extWEAK simulations. The coldest area-averaged SST of 285.12 K is found in extWEAK (Tab. 1).

### 2.3 Warm conveyor belt trajectories

We employ the Lagrangian perspective to identify the WCB airstream as a coherent ensemble of strongly ascending trajectories (Wernli, 1997; Madonna et al., 2014). Specifically, 48 h forward trajectories are computed every hour between 18 and 25 February 2019 with *LAGRANTO* (Sprenger and Wernli, 2015). Trajectories are started from a 50 km equidistant horizontal grid spanning the North Atlantic region between 80 to 25 ° W and 30 to 55 ° N. They are initiated from 10 chosen vertical levels within the lower-most 2 km, specifically at altitudes of 50, 200, 400, 600, 800, 1000, 1250, 1500, 1750, and 2000 m. The following variables are traced along the trajectories: pressure height, temperature, specific humidity, and potential vorticity.



From all 48 h trajectories, WCB trajectories are selected as those that ascent at least 500 hPa. To avoid double-counting of trajectories, a filter following the approach of Madonna et al. (2014) is applied. In total, approximately 10 % of all 40 710 trajectories are identified as WCB trajectories. We categorize WCB trajectories into three distinct stages based on their pressure ($p$, e.g., Schäfler et al., 2014; Quinting et al., 2022): lower-tropospheric inflow region ($p \geq 800$ hPa), ascent region (800 hPa$>$ $p >400$ hPa), and the outflow region in the upper troposphere ($p \leq 400$ hPa). Using this stratification, we re-grid the positions of WCB air parcels onto the Eulerian grid, resulting in consistent hourly masks for WCB inflow, ascent, and outflow, respectively. Furthermore, we apply the nearest neighbor method, as described by Škerlak et al. (2014), to also re-grid potential temperature, specific humidity, and potential vorticity traced along the WCB trajectories for inflow, ascent, and outflow, respectively.

## 3   Case study introduction

In the following, we introduce the nine-day case study characterized by the passage of multiple cyclones across the Gulf Stream region and downstream ridge amplification and discuss its representation in the CNTRL and IDEA simulations. In February 2019, Great Britain and Western Europe encountered unprecedented warmth, with temperatures reaching 10 to 15 K above the climatological average (Kendon et al., 2020). The exceptional temperature anomalies over Europe in February 2019 were caused by a combination of various processes. Crucially, the development of an upper-level ridge extending from northwest Africa to southern Scandinavia, along with anticyclonic circulation, enabled the southwestward movement of warm, maritime air masses into Western Europe (Young and Galvin, 2020). Spanning from 20 to 27 February, the event is categorized as a European Blocking event according to the seven-regime classification (Grams et al., 2017). This is characterized by a positive 500 hPa geopotential height anomaly over the Eastern North Atlantic and a concurrent negative anomaly centered over Greenland. The atmospheric blocking was preceded and accompanied by the development of several, rapidly intensifying cyclones. Wenta et al. (2024) suggested that those cyclones originating from the North Atlantic provided conditions for both the moistening of air masses over the Gulf Stream and their ascent into the upper troposphere, potentially contributing to the maintenance and amplification of the block.

Our analysis focuses on the synoptic conditions during the evolution of the two most intensive cyclones as well as the subsequent amplification of the upper-level ridge and the blocking onset, which are covered by the nine-day simulation period.

In the following, the synoptic evolution is presented using the results from the ICON CNTRL simulation. Between 18 UTC 18 February 2019, and 00 UTC 20 February 2019, the first cyclone (CY1) moved along the SST gradient in the Gulf Stream region, beginning its intensification at 00 UTC 19 February (Fig. 2a,c). The cyclone then progressed northward by 00 UTC 20 February. During CY1's passage across the North Atlantic, cold continental air was advected over the warm waters south of the Gulf Stream, triggering a CAO (Fig. 2b). This CAO over the warm Gulf Stream waters resulted in significant heat and moisture exchange between the ocean and the atmosphere (see also Wenta et al., 2024). The WCB trajectories show that the ascent associated with this blocking event, especially at 12 UTC 19 February, occurred in the Central North Atlantic, with an inflow from the Western part of the North Atlantic. The outflow of WCB trajectories associated with CY1 (Fig. 2c) was located over the Eastern North Atlantic, in the developing ridge, as indicated by the PV distribution at 315 K (Fig. 2c).



On 21 February, a second cyclone (CY2), intensified rapidly in the Gulf Stream region (Fig. 2d-f). Over a period of two days, as it crossed the North Atlantic, it significantly strengthened and triggered another CAO event (Fig. 2e). CY2 moved into the area where CY1 had previously undergone rapid intensification, impacting the atmospheric boundary layer through moistening and heating. This sequential pattern of CY2 following CY1's track, suggests a cyclone-cyclone preconditioning mechanism, during which moisture from the region behind cyclone CY1 is fed into the warm sector of the subsequent cyclone and its associated WCB (Fig. 2f; Papritz et al., 2021; Demirdjian et al., 2023). Indeed, Wenta et al. (2024) demonstrated that a significant portion of the moisture in the ascending WCB airstreams of the February 2019 cyclones originates locally and is linked to the preceding cyclone activity. Similar to CY1, the outflow of CY2 is primarily situated within the established ridge over Western Europe, with the upstream WCB ascent predominantly taking place in the Central North Atlantic (Fig. 2f). On 24 February, the CY2 WCB trajectories further reinforce the stationary ridge and support its eastward expansion (Fig. 2i).

The synoptic evolution in the ICON simulations, specifically CNTRL and IDEA (Fig. A1 and Christ, 2023, for details), compares well with the ERA5 re-analysis (Hersbach et al., 2020). Overall, the CNTRL and IDEA simulations are very similar and only small differences in sea level pressure (SLP) and upper-level PV emerge, and subsequently grow with increasing lead time. All key synoptic features that lead to the onset of blocking are consistently depicted in the both experiments, and no notable initial divergence is observed in any of the experiments with modified SST conditions. Instead, differences between the experiments gradually emerge and intensify as the lead time increases. In the next section, we describe the results of our comprehensive analysis of sensitivity experiments, each incorporating different SST configurations in the Gulf Stream region (Section 2.2) and focus on the impacts on air-sea interactions (Section 4.1), the moistening and heating of the overlying atmosphere (Section 4.2), cyclone evolution (Section 4.3), WCB ascent (Section 4.4), and finally the evolution of the upper-level ridge (Section 4.5).

## 4 The role of SST perturbations in the Gulf Stream region

Our analysis employs both Eulerian and Lagrangian perspectives. The Eulerian perspective focuses on the evolution of atmospheric conditions within the Gulf Stream region during the development of cyclones CY1 and CY2, whereas the Lagrangian perspective focuses on the evolution and properties of WCB trajectories associated with CY1 and CY2 that connect the lower and upper troposphere. As noted above, the differences between IDEA and CNTRL are small and we find a very similar synoptic evolution in IDEA and CNTRL, which includes cyclone tracks, WCB ascent, and the upper-level flow evolution. For this reason, IDEA is taken as a reference for the comparison of the sensitivity experiments, and SST impacts are quantified as the difference between IDEA and any other experiment. Using CNTRL instead of IDEA as a reference would yield qualitatively similar results.

### 4.1 Impact on air-sea interactions

The impact of SST perturbations on air-sea interactions is investigated through spatially averaged surface heat fluxes in the Gulf Stream region, spanning from 30 to 55° N and 80 to 25° W. An analysis of the temporal changes in latent and sensible heat



**Figure 2.** SST, CAOs and PV at 315 K during the transition of cyclones CY1 and CY2 across the North Atlantic in February 2019. **(a,d,g)** Sea level pressure (grey contours, every 5 hPa), 2-PVU potential vorticity (PV) contour at 315 K (black bold), and sea surface temperature (SST, shading, in K) for **(a)** 12 UTC 19 February 2019, **(c)** 12 UTC 22 February 2019, and **(e)** 12 UTC 24 February 2019. **(b,e,h)** CAO index, defined as $\theta_{SST} - \theta_{850}$ (blue shading in K), sea level pressure (grey contours, every 5 hPa), 2-PVU potential vorticity (PV) contour at 315 K (black bold) at the same times shown in **(a,c,e)**. **(c,f,i)** PV at 315 K (shading in PVU), WCB outflow masks (grey shading) at the same times shown in **(a,c,e)**, and WCB trajectories initialized 24 h prior to the time shown (in hPa). All panels show the CNTRL simulation (see Section 2.1).

fluxes reveals a similar evolution of the surface fluxes across the different experiments (Fig. 3). First of all, the removal of the

small-scale variability associated with individual Gulf Stream meanders in IDEA does not translate into notable differences in spatially averaged surface latent and sensible heat fluxes compared to CNTRL (Tab. 2 and Fig. 3). More pronounced differences arise for the other experiments. The P1.5K experiment consistently shows the highest upward surface fluxes from the ocean to





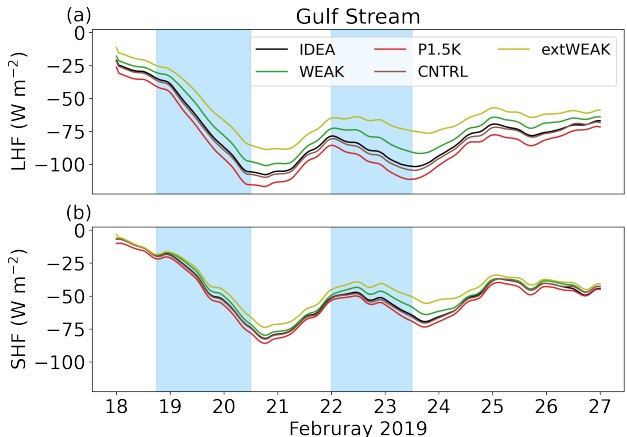

**Figure 3. (a)** Latent heat flux (LHF, in $W\,m^{-2}$, upward defined negatively) spatially averaged over the Gulf Stream region (30 to 55 °N and 80 to 25 °W) for CNTRL (brown), IDEA (black), WEAK (green), extWEAK (light) green, and P1.5K (red). **(b)** as **(a)** but for sensible heat flux (SHF). Blue shading outlines the time period of the passage of cyclone 1 (CY1) and cyclone 2 (CY2) in the Gulf Stream region.

the atmosphere, in contrast the extWEAK experiment displays the lowest fluxes. This difference is particularly pronounced for latent heat flux (Fig. 3a and Tab. 2) and also applies to a lesser extent to the sensible heat flux (Fig. 3b). Specifically, a latent heat

fluxes increase of up to $10\,W\,m^{-2}$ for P1.5K and a decrease of $25\,W\,m^{-2}$ for extWEAK compared to IDEA is observed in the Gulf Stream region. Sensible heat fluxes differences of up to $5\,W\,m^{-2}$ in P1.5K and up to $20\,W\,m^{-2}$ in extWEAK compared to IDEA (Fig. 3b) are present. These differences are particularly pronounced during the passage of cyclones CY1 and CY2, which trigger CAO events in the Western North Atlantic (Fig. 2b,e). During these events, the exchange of heat and moisture between the ocean and atmosphere intensifies, resulting from the large contrast between the colder and dry air following the cyclones

and the warmer SST. From a temporally averaged perspective (not shown), the P1.5K experiment, with its warmed SST, shows the largest increase in mean upward surface heat fluxes around the Gulf Stream SST front. Conversely, in experiments with weakened SST gradients (WEAK and extWEAK), we observe a decrease in surface heat fluxes, particularly south of the SST front. These findings align with prior research indicating reduced upward surface fluxes on the warmer, southern side of the SST gradient is weakened (Vries et al., 2019; Tsopouridis et al., 2021).

**4.2 Impact on air temperature and specific humidity**

The changes in sensible and latent heat fluxes impact the air temperature and specific humidity within the atmospheric boundary layer over the Gulf Stream region. Table 2 outlines the average differences in air temperature and specific humidity at 925 hPa in the Gulf Stream region between IDEA and each experiment. The removal of the small-scale variability of the Gulf Stream meanders in IDEA does not translate into notable differences in surface heat fluxes (Tab. 2 and Fig. 3). Consequently, the

differences in area-averaged temperature and specific humidity between CNTRL and IDEA are relatively small throughout the simulation period. Consistent with the largest changes in surface fluxes, the largest differences to IDEA are observed in





**Table 2.** Differences between experiments for surface latent and sensible heat fluxes, as well as air temperature, specific humidity, and baroclinicity at 925 hPa. Differences are calculated relative to the idealized SST gradient experiment (IDEA), and are temporally averaged over the whole simulation period (19 to 27 February 2019) and spatially averaged across the Gulf Stream region (30 to 55 °N and 80 to 25 °W).

|  | latent heat flux | sensible heat flux | air temperature | specific humidity | baroclinicity |
|---|---|---|---|---|---|
| IDEA−CNTRL | $1.83\pm15.21\,\mathrm{W\,m^{-2}}$ | $0.39\pm9.90\,\mathrm{W\,m^{-2}}$ | $-0.06\pm0.5\,\mathrm{K}$ | $-0.02\pm0.29\,\mathrm{g\,kg^{-1}}$ | $-0.0004\,\mathrm{K\,km^{-1}}$ |
|  | -2.36% | -0.80% | -0.02% | -0.67% | -0.9% |
| IDEA−WEAK | $-6.64\pm21.01\,\mathrm{W\,m^{-2}}$ | $-2.41\pm14.75\,\mathrm{W\,m^{-2}}$ | $0.01\pm0.88\,\mathrm{K}$ | $0.06\pm0.44\,\mathrm{g\,kg^{-1}}$ | $0.001\,\mathrm{K\,km^{-1}}$ |
|  | 8.56% | 4.95% | 0.0% | 2.18% | 2.31% |
| IDEA−extWEAK | $-15.95\pm33.5\,\mathrm{W\,m^{-2}}$ | $-6.26\pm23.04\,\mathrm{W\,m^{-2}}$ | $0.04\pm1.62\,\mathrm{K}$ | $0.16\pm0.82\,\mathrm{g\,kg^{-1}}$ | $0.003\,\mathrm{K\,km^{-1}}$ |
|  | 20.6% | 12.83% | 0.02% | 5.24% | 7.07% |
| IDEA−P1.5K | $7.42\pm10.54\,\mathrm{W\,m^{-2}}$ | $2.9\pm6.77\,\mathrm{W\,m^{-2}}$ | $-0.32\pm0.58\,\mathrm{K}$ | $-0.09\pm0.36\,\mathrm{g\,kg^{-1}}$ | $-0.0006\,\mathrm{K\,km^{-1}}$ |
|  | -9.58% | -5.95% | -0.12% | -3.16% | -1.43% |

the P1.5K experiment, with an average air temperature increase of 0.32 K and specific humidity increased by $0.09\,\mathrm{g\,kg^{-1}}$. Conversely, the WEAK and extWEAK experiments show a decrease in air temperature (by 0.01 and 0.04 K, respectively) and specific humidity (by 0.06 and $0.16\,\mathrm{g\,kg^{-1}}$, respectively).

In light of our study's focus on WCBs, we further evaluate the variations in air temperature and specific humidity specifically within the WCB inflow region. This region is primarily located south of the SST front, where SSTs are relatively warm. This region is important for moisture supply for subsequent latent heat release during WCB ascent (Wenta et al., 2024). Here, the Eulerian WCB inflow region is determined based on the positions of WCB air parcels in the IDEA experiment, and specifically defined as locations where the trajectories' pressure exceeds 800 hPa. Subsequently, the inflow region is confined to areas

where the WCB inflow is consistently present for at least 30% of the nine-day simulation (Fig. A3). Notably, the location of this Eulerian WCB inflow region remains consistent across different experiments, owing to the similar synoptic evolution across all experiments. In the WCB inflow region, the evolution of the spatially averaged vertical profiles for air temperature and specific humidity show gradually evolving differences between experiments (Figs. 4 and 5). In the WEAK and more notably in the extWEAK experiments, air temperature is generally slightly lower than in IDEA (Fig. 4b,c) with a simultaneous

reduction in moisture content (Fig. 5b,c). Initially, temperature differences are confined to the lower troposphere but throughout the simulation differences up to the tropopause level arise. The difference is particularly striking in the extWEAK experiment, where the weakening of the SST gradient reduces the specific humidity from the surface up to 500 hPa several days into the simulation (Fig. 5c) with a concomitant decrease in temperature throughout the entire troposphere (Fig. 4c). In contrast, the P1.5K experiment shows an increase in both air temperature (Fig. 4d) and moisture content up to 700 hPa (Fig. 5d), in line with

the previously discussed increase in surface sensible and latent heat fluxes (Section 4.1). The temperature differences between all experiments are particularly pronounced after the passages of cyclones CY1 and CY2 (Fig. 4) and extend from the boundary layer into the upper troposphere. Overall, the spatio-temporal evolution of temperature differences in the experiments illustrates





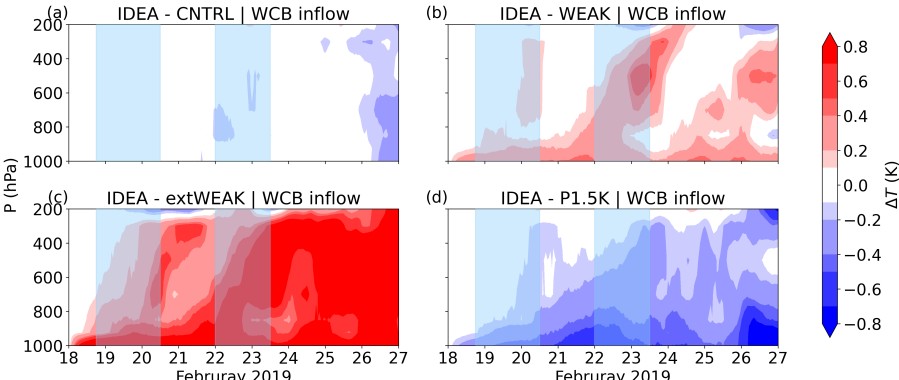

**Figure 4. (a)-(d)** Evolution of vertical profiles of air temperature differences ($\Delta$ T, shading, in K) spatially averaged in the Eulerian WCB inflow region defined as the area where the WCB inflow frequency in IDEA (WCB trajectories' pressure larger than 800 hPa) exceeds 30 % during the nine-day simulation period. Differences are shown for **(a)** IDEA-CNTRL, **(b)** IDEA-WEAK, **(c)** IDEA-extWEAK, and **(d)** IDEA-P1.5K. Light Blue shading outlines the time period related to the passage of cyclone CY1 and cyclone 2 CY2 in the Gulf Stream region.

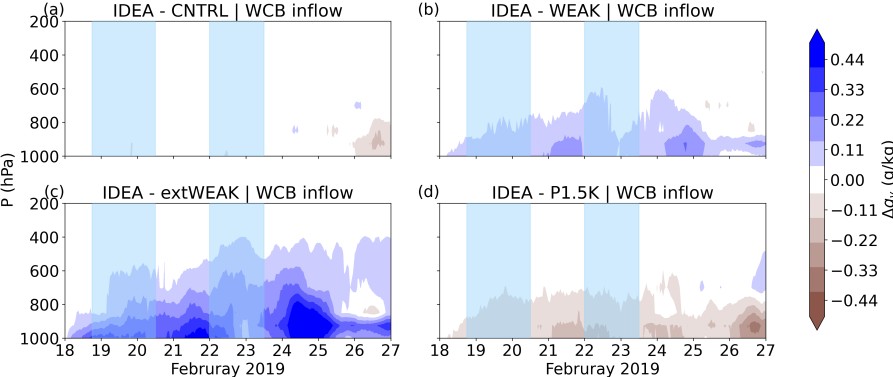

**Figure 5.** As Fig 4 but for specific humidity differences ($\Delta$ q$_v$, shading, in g kg$^{-1}$).

the influence of the prescribed SST perturbations on the atmosphere, whereby differences evolve from the lower troposphere and subsequently progress upward after the passages of CY1 and CY2.

The evolution of temperature and humidity profiles described above is related to altered vertical motion coupled with changes in diabatic heating from cloud formation processes during WCB ascent. The availability of moisture in the WCB inflow region determines the degree of latent heat released during cloud formation, and thus, the cross-isentropic WCB ascent strength (e.g., Oertel et al., 2023a). First of all, we use accumulated precipitation in the North Atlantic (30 to 60° N and 80 to 0° W) as a proxy for the mass conversion of water vapor to the liquid/solid state. Throughout the nine-day period the accumulated precipitation

differs between the experiments (Fig. 6). The highest precipitation sums, and approximately 6% more precipitation than in IDEA, occur in the P1.5K experiment, which suggests an enhancement in diabatic heating of the mid-troposphere during WCB



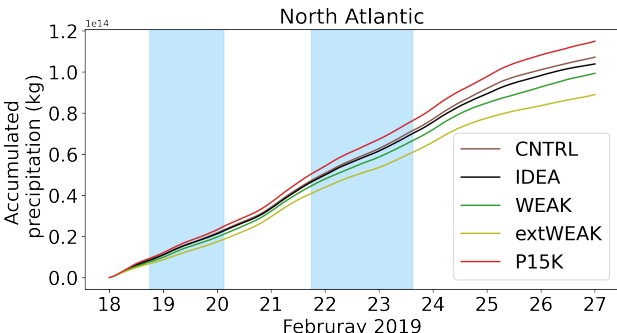

**Figure 6.** Evolution of spatiotemporally accumulated surface precipitation sums (in kg) in the North Atlantic (30 to 60 $^\circ$ N and 80 to 0 $^\circ$ W) during the nine-day simulation period for CNTRL (brown), IDEA (black), WEAK (green), extWEAK (light green), and P1.5K (red). Blue shading outlines the time of the passage of cyclone 1 (CY1) and cyclone 2 (CY2) in the Gulf Stream region.

ascent. Conversely, experiments with weaker SST gradients, specifically WEAK and extWEAK, accumulate 5% and 16% less precipitation, respectively, than IDEA, which, in turn, is associated with reduced diabatic heating.

The variability in precipitation sums suggests that differences in lower tropospheric moisture content across experiments noticeably influence diabatic heating rates from parameterized cloud microphysical processes and convection. The analysis of diabatic heating tendencies from the individual model parameterizations (Oertel et al., 2023a) confirms that cloud microphysics and the convection scheme substantially influence total heating and also differ between the experiments (Figs. A4, A5 and A6, and detailed discription in Christ, 2023). Consistent with the differences in surface precipitation, the P1.5K experiment shows enhanced cloud microphysical diabatic heating rates above 3-4 km altitude (Fig. A4c), whereas the heating rates in the WEAK

and extWEAK experiments are reduced (Fig. A4b,d). Similarly, the convection parameterization (Fig. A6), which is most active in the lowest 2 km, is more active in the P1.5K experiment and reduced diabatic heating arises in the WEAK and extWEAK experiments. This indicates that temperature differences propagate from the sea surface into the boundary layer, where the WCB airstream picks up additional moisture and subsequently warms the mid-troposphere during ascent through latent heating, efficiently redistributing temperature and humidity. To conclude, diabatic heating during WCB ascent and associated surface

precipitation sums are affected by the availability of moisture in the WCB inflow region, which in turn is influenced by SST perturbations in the Gulf Stream region. In Section 4.4, we elaborate in more detail on the impact on WCB ascent.

## 4.3    Impacts on cyclones

Building upon the discussions above, cyclones CY1 and CY2 and their associated WCBs play a critical role in amplifying and transmitting the influence of SST into higher atmospheric layers. Generally, the genesis of cyclones is linked to baroclinicity

(Charney, 1947), which provides information on the potential energy available for cyclones. We calculate baroclinicity as the horizontal gradient of equivalent potential temperature at the 925 hPa level. This brings into focus the potential impact of the strength of the SST gradient on cyclone development through changes in low-level baroclinicity in the Gulf Stream region.





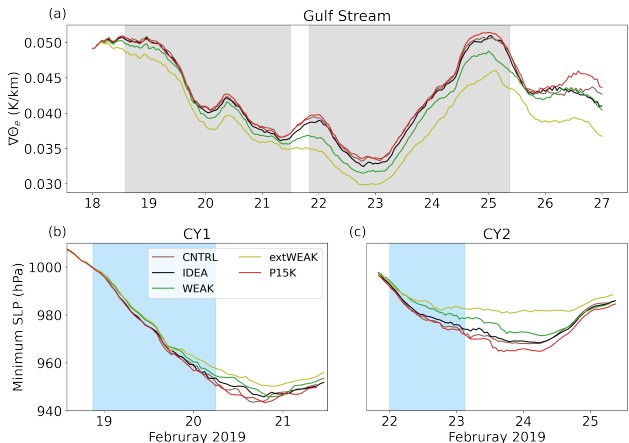

**Figure 7. (a)** Evolution of baroclinicity at 925 hPa ($\nabla\Theta_e$, in $\mathrm{K\,km^{-1}}$) for CNTRL (brown), IDEA (black), WEAK (green), extWEAK (light green), and P1.5K (red) spatially averaged over the Gulf Stream region (30 to 55 ° N and 80 to 25 ° W). **(b,c)** Evolution of minimum sea level pressure (SLP) in the cyclone center (in hPa) for **(b)** the first cyclone (CY1) and **(c)** the second cyclone (CY2) for the experiments shown in **(a)**. Blue shading outlines the time of the passage of cyclone 1 (CY1) and cyclone 2 (CY2) in the Gulf Stream region. Grey shading in **(a)** indicates the times of CY1 and CY2 shown in **(b)** and **(c)**, respectively.

Spatial averages of low-level baroclinicity in the Gulf Stream region (Fig. 7a, Tab. 2) demonstrate a reduction of baroclinicity in the WEAK and notably the extWEAK experiments in comparison to IDEA with on average 2.3% and 7% reduction,

respectively. Despite an identical SST gradient, the P1.5K experiment shows a slight increase in baroclinicity compared to IDEA, which can be attributed to differential heating and/or advection. In contrast, for spatial averages, only minor differences between CNTRL and IDEA experiments are present, although for temporal averages locally larger differences arise, reflecting the patterns of surface heat flux differences. This suggests that very small-scale SST meanders have only a small effect on baroclinicity in our experiments, and thus, also hardly influence cyclone development.

Yet, experiments with modified baroclinicity and moisture availability resulting from changes in air-sea interactions, show different cyclone development. The analysis of cyclone tracks for CY1 and CY2 illustrates that while the cyclones' positions (Fig. A2) and their peak intensity time (at 21 UTC 20 February 2019 for CY1 and at 00 UTC 24 February 2019 for CY2) remain consistent across different experiments, their intensity notably varies. The intensity is measured by the minimum SLP at the cyclone center (Fig. 7b,c). CY1 deepens more strongly in CNTRL and P1.5K and minimum SLP drops to 943 hPa in

both experiments, compared to the IDEA with a minimum SLP of 945 hPa (Fig. 7b). A slight reduction in deepening rates is observed for the WEAK and extWEAK experiments which reach an SLP minimum of 945 and 950 hPa respectively. Cyclone intensification differences between experiments are substantially larger for CY2 (Fig. 7c) than for CY1 for which differences on the order of 15 hPa occur. CY2 in WEAK has a higher SLP (minimum of 971 hPa) than IDEA, and the minimum SLP in extWEAK is markedly higher (minimum of 980 hPa). Conversely, the P1.5K cyclone demonstrates the most rapid and strongest

deepening to a minimum of 964 hPa. Meanwhile, the evolution of minimum SLP in CNTRL and IDEA is relatively similar with





a minimum of 968 hPa, i.e., SLP minimum differences for CY2 reach about 2 hPa. The stronger effect of SST perturbations on CY2 suggests that the impact of modified air-sea interactions on the synoptic evolution increases with lead time, likely also influenced by strong surface fluxes associated with the CAO following CY1. Overall, the reduction of cyclone intensity in experiments with weaker SST gradients (WEAK and extWEAK) aligns with the decreased low-level baroclinicity and reduced diabatic heating (Section 4.2).

## 4.4 Impact on WCB ascent

The deep ascent of WCB airstreams through the entire atmospheric column can connect low-level processes and upper-level flow patterns. To gain insight into the relationship between SST in the Gulf Stream region and WCB ascent, we employ a Lagrangian approach to objectively characterize WCB properties. The WCBs associated with cyclones CY1 and CY2 ascend poleward from the warm sector of the cyclones ahead of the cold front into the upper-level ridge (see, e.g., Fig. 2i). The typical ascent characteristics of identified WCBs are depicted for the trajectories from IDEA (Fig. 8). The average ascent of WCB air parcels starts in the inflow layer, below 800 hPa and ends in the outflow layer, above 400 hPa (Fig. 8a). Due to cloud formation processes during ascent, specific humidity continuously decreases from $\sim7$ to $\sim0$ g kg$^{-1}$ (Fig. 8b). The increase in liquid and ice water contents (Fig. 8c) during the ascent signifies the onset of cloud formation processes. Latent heat release, primarily from condensation and depositional growth of ice hydrometeors (e.g., Oertel et al., 2023a), enables the cross-isentropic ascent and increases the WCB air parcels' potential temperature from, on average, 289 to 305 K (Fig. 8d). The increase in potential temperature resulting from latent heating is linked to a modification of PV. PV is generated below the level of maximum latent heat release and reduced above (Fig. 8e), leading to low PV values in the upper tropospheric WCB outflow region (Fig. 11c) and aligning with the conceptual model of PV changes along WCB ascent (Hoskins et al., 1985; Wernli, 1997; Madonna et al., 2014)

In the subsequent sections, we will explore the differences in WCB trajectory characteristics between the five SST experiments (Tab. 3). Fig. 9 displays the number of WCB trajectories starting their ascent every hour throughout the simulations. Of the total 40 710 trajectories that were started every hour, approximately 2000 to 6000 are identified as WCB trajectories depending on the experiment. The two local maxima in WCB trajectory number on 19 and 22 February are associated with the passages of cyclones CY1 (18 to 20 February 2019) and CY2 (22 to 23 February) across the Gulf Stream region (blue shading in Fig. 9). The overall evolution of WCB trajectory starts is similar across all experiments, however, the difference in absolute WCB trajectory numbers, and thus, mass transport across the troposphere, is considerably large, and relative differences of up to 20-50 % emerge between experiments (Fig. 9b, Tab. 3). An up to 20 % increase in WCB trajectory numbers in P1.5K compared to IDEA indicates a stronger WCB ascent associated with enhanced mass flux. Contrarily, up to 50 % less WCB trajectories are present in extWEAK (Fig. 9b) indicating substantially weakened WCB ascent. Similarly, the WCB trajectory numbers of WEAK are reduced by up to 20 % compared to IDEA. Overall, the difference in the number of WCB trajectories between the IDEA and CNTRL experiments is relatively small. However, differences of up to 20 % occur occasionally, generally showing a tendency for higher numbers in the CNTRL experiment. On average the largest number of WCB trajectories is



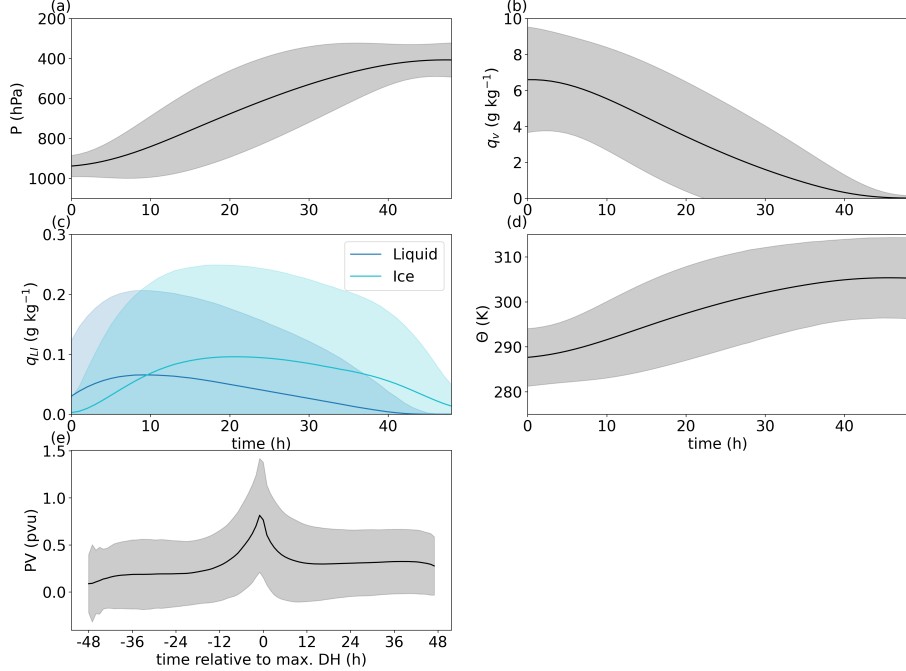

**Figure 8.** Mean evolution of **(a)** pressure ($p$, in hPa), **(b)** specific humidity ($q_v$, in g kg$^{-1}$), **(c)** potential temperature ($\Theta$, in K), **(d)** liquid and ice hydrometeor contents ($q_{LI}$, blue and light blue lines, in g kg$^{-1}$), and **(e)** potential vorticity (PV, in PVU) along WCB trajectories. The PV evolution in **(e)** is centered relative to maximum diabatic heating (DH). The shading shows the mean $\pm$ standard deviation. All figures are shown for the IDEA WCB trajectories.

present in P1.5K (Tab. 3) with 9.4 % more WCB trajectories than in IDEA. A reduction of SST gradient reduces the numbers
385 of WCB trajectories by on average -9.43 % in WEAK and -30.07 % in extWEAK.

Ascent characteristics of WCB trajectories are often described by the distribution of their ascent times (e.g., Oertel et al., 2021). The distribution of ascent timescales reveals that the typical duration, in this case, lies within 9 to 20 h (Fig. 9c). The comparison across all experiments shows a shift in ascent timescale distribution (Fig. 9c, Tab. 3). Notably, in the extWEAK experiment, the distribution of ascent timescales shifts towards longer durations, indicating slower ascent (Fig. 9c). Specifically,
390 the frequency of ascent timescales below 18 hours is substantially reduced (Fig. 9c). In the WEAK experiment, rapid ascent with timescales below 10 hours occurs less frequently, whereas in the P1.5K experiment, rapid ascent is more frequent. To summarize, simulations with a reduced SST gradient are characterized by weaker WCB ascent and a lower frequency of fast ascents. Conversely, higher SST in P1.5K results in an increased number of WCB trajectories, especially in a larger number of rapidly ascending WCB trajectories. Finally, no substantial differences between IDEA and CNTRL arise in this context.
395 In addition to differences in the number of WCB trajectories and ascent timescale distributions, other WCB properties are influenced by the SST perturbations (Tab. 3). These differences are evident not only from the Lagrangian perspective but also from the Eulerian viewpoint, which was discussed in Section 4.2. As shown above, WEAK and extWEAK are characterized by





**Table 3.** Differences in WCB characteristics between the five experiments. Shown are temporally averaged absolute and relative warm conveyor belt (WCB) trajectory numbers as well as mean 500 hPa ascent times ($\tau_{500}$), absolute pressure change during WCB trajectories' ascent ($\Delta p$), absolute specific humidity change ($\Delta q_v$), absolute change of potential temperature ($\Delta \Theta$), trajectories' mean minimum pressure (min. p) and mean maximum potential temperature (max. $\Theta$). Differences are calculated relative to the idealized SST gradient experiment (IDEA) and averaged over all WCB trajectories for the entire period (19 to 25 February 2019).

| | WCB number | $\tau_{500}$ | abs. $\Delta p$ | $\Delta q_v$ | $\Delta \Theta$ | min. p | max. $\Theta$ |
|---|---|---|---|---|---|---|---|
| IDEA−CNTRL | -124 | -0.19 h | 0.51 hPa | -0.06 g kg$^{-1}$ | -0.19 K | -1.68 hPa | -0.19 K |
| | -4.03% | -0.91% | 0.09% | -0.87% | -0.99% | -0.45% | -0.06% |
| IDEA−WEAK | 291 | -0.63 h | 2.48 hPa | 0.30 g kg$^{-1}$ | 0.80 K | -2.90 hPa | 1.33 K |
| | 9.43% | -3.02% | 0.43% | 4.11% | 4.21% | -0.78% | 0.44% |
| IDEA−extWEAK | 927 | -2.54 h | 9.98 hPa | 0.74 g kg$^{-1}$ | 2.20 K | -11.65 hPa | 3.57 K |
| | 30.07% | -12.16% | 1.73% | 10.10% | 11.62% | -3.13% | 1.17% |
| IDEA−P1.5K | -290 | 0.22 h | -2.35 hPa | -0.17 g kg$^{-1}$ | -0.52 K | 1.95 hPa | -0.79 K |
| | -9.40% | 1.05% | -0.41% | -2.32% | -2.72% | 0.52% | -0.26% |

reduced moisture and heat supply to the WCB inflow region due to altered air-sea interactions, and vice versa for P1.5K. This results in differences in specific humidity as well as potential temperature and pressure changes during the WCB ascent among the experiments (Fig. 10, Tab. 3). Due to larger specific humidity in the WCB inflow, WCB ascent in P1.5K is associated with the largest average loss of specific humidity of 9 g kg$^{-1}$ (Fig. 10b), which translates to the largest diabatic heating of approximately 22 K (Fig. 10c). The slightly lower specific humidity loss in WEAK and extWEAK of on average 8.75 and 8.4 g kg$^{-1}$ (Fig. 10b) consequently leads to a reduced diabatic heating of 21 and 19 K, respectively (Fig. 10c).

Consistent with larger moisture loss and diabatic heating, P1.5K shows the deepest ascent quantified by average absolute pressure change of 595 hPa (Fig. 10a). The WCB trajectory ascent in the WEAK and extWEAK is weaker than in IDEA with pressure differences of on average 585 and 575 hPa, while the differences between IDEA and CNTRL are negligible.

Overall, the average differences in diabatic heating (Figs. 6 and 10c), as well as ascent strength among the experiments (Fig. 10) are consistent throughout the analysis period. Differences in specific humidity and heat supply to the WCB inflow region in response to changed air-sea interactions lead to different WCB characteristics and diabatic heating during ascent. Intuitively, reduced specific humidity loss during WCB ascent in the weaker SST gradient experiments results in lower diabatic heating and weaker ascent. Vice versa, increased surface fluxes from higher SST in the P1.5K experiment lead to faster and more pronounced WCB ascent.

The altered diabatic processes during the WCB ascent also influence the WCB characteristics in the upper troposphere (Tab. 3). On average, WCB air parcels in the P1.5K experiment reach 320 hPa, which is very close to the average pressure in CNTRL and IDEA of 318 hPa for both experiments. In contrast, in WEAK and extWEAK, average WCB outflow height is lower with mean minimum pressure values of ∼330 and 345 hPa, respectively (Fig. 11a). Furthermore, differences in average potential temperature in the WCB outflow, i.e. maximum isentropic level, are present and vary between 2 and 10 K (Fig. 11b, Tab.



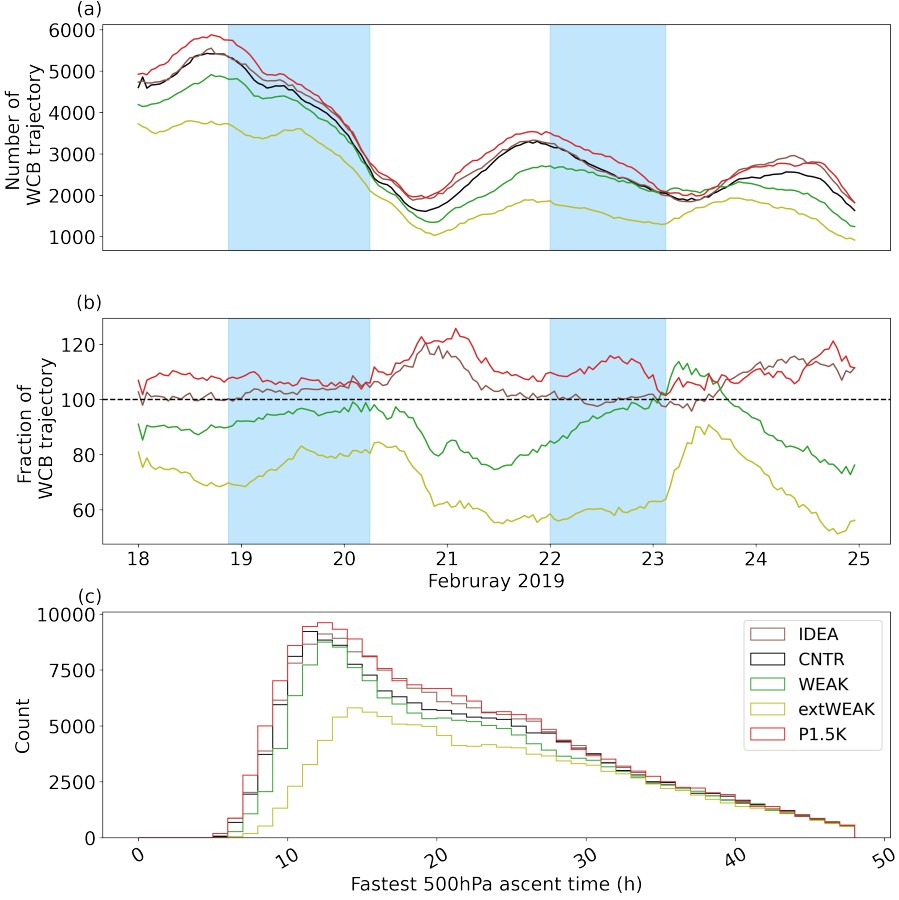

**Figure 9. (a)** Number of WCB trajectories starting between 18 to 25 February 2019 for IDEA (black), WEAK (green), extWEAK (light green), and P1.5K (red). The time axis represents the time of the start of the 2 day WCB trajectory ascent. **(b)** as **(a)** but for the WCB trajectory number fraction relative to IDEA (in %). Blue shading in **(a,b)** outlines the time according to the passage of cyclone 1 (CY1) and cyclone 2 (CY2) in the Gulf Stream region. **(c)** Histograms of the fastest 500 hPa ascent times for all WCB trajectories for the five experiments (bin width 1 h), colors as in **(a,b)**.

3). This aligns with differences in diabatic heating: Throughout the simulation reduced cross-isentropic ascent in WEAK and extWEAK compared to IDEA and CNTRL results in lower maximum potential temperature in the WCB outflow, while WCB
trajectories in P1.5K reach slightly higher isentropic levels. Although differences in WCB outflow potential temperature are present, PV values averaged in the WCB outflow (defined as all WCB trajectory positions with pressure values lower than 400 hPa) do not vary substantially between the experiments and amount to on average 0.2 PVU (Fig. 11c).





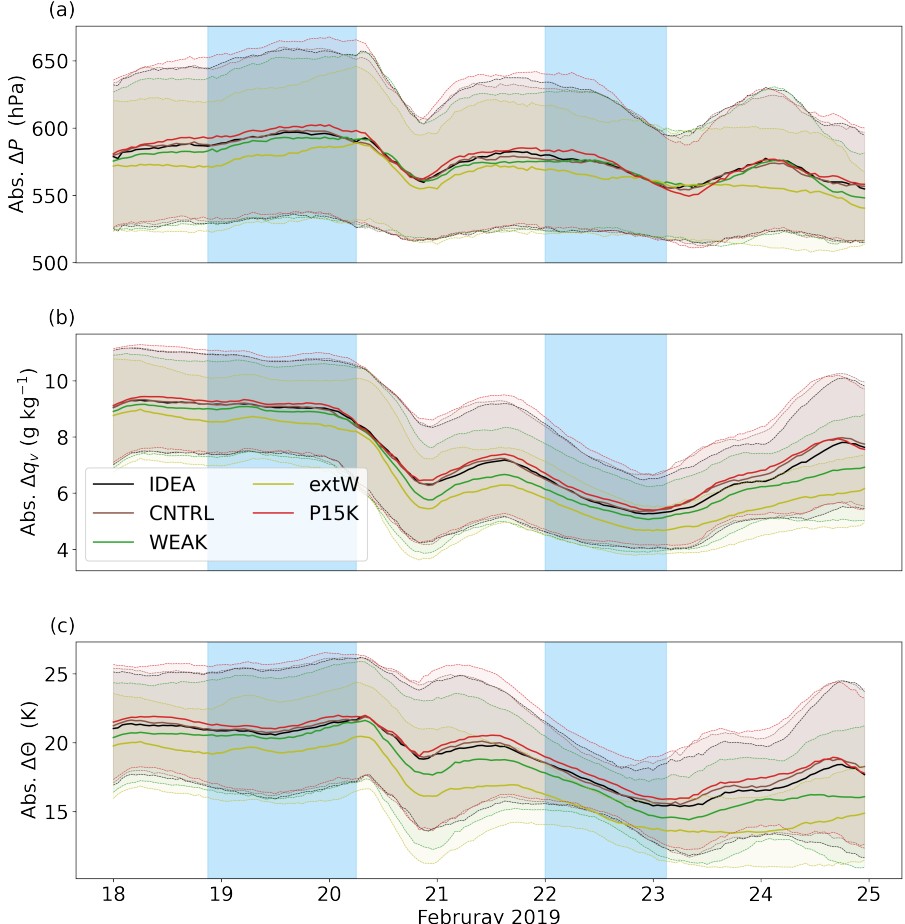

**Figure 10. (a)** Evolution of mean absolute pressure change during WCB trajectories' ascent ($\Delta p$, in hPa, the date indicates the start time of trajectories) for CNTRL (brown), IDEA (black), WEAK (green), extWEAK (light green), and P1.5K (red) including mean $\pm$ standard deviation (shading). **(b)** as **(a)** but for specific humidity change ($\Delta q_v$, in g kg$^{-1}$. **(c)** as **(a)** but for absolute change of potential temperature ($\Delta \Theta$, in K). Blue shading outlines the time period of the passage of cyclone 1 (CY1) and cyclone 2 (CY2) in the Gulf Stream region.

### 4.5 Impact on the large scale flow

In all experiments, the WCB outflow predominantly aligns with the intensifying upper-level ridge, contributing to the formation

of the quasi-stationary anticyclone associated with the European blocking. Fig. 12 illustrates the impacts of modified WCB outflow properties on the upper-tropospheric ridge characteristics by re-gridding WCB trajectory outflow positions to the Eulerian grid. The WCB outflow region is located near the Western coast of Europe downstream of the Gulf Stream region. Only small spatial differences in WCB outflow location between experiments are present, which is consistent with an overall similar large-scale flow evolution. Yet, the northern boundary of the WCB outflow in WEAK and extWEAK is located slightly





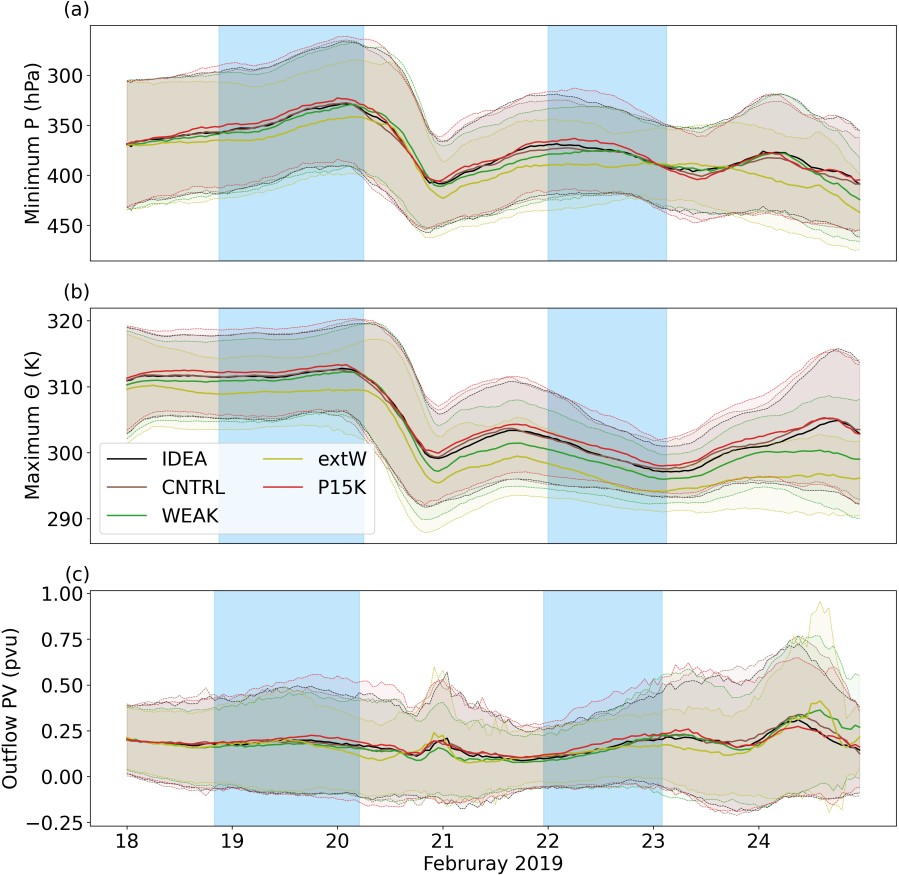

**Figure 11.** As Fig. 10 but for **(a)** mean minimum pressure (*p*) of WCB trajectories (in hPa), **(b)** mean maximum potential temperature (Θ) of WCB trajectories (in K), and **(c)** mean potential vorticity (PV) of WCB trajectories outflow (in pvu).

further south than in IDEA (Fig. 12b,d). In both experiments, the position of the upper-level jet is also displaced further south. We hypothesize that the weaker divergent outflow of WCBs in WEAK and extWEAK influences the amplification of the downstream ridge. A consistent change in WCB outflow location is present in P1.5K. Specifically, the WCB outflow is slightly shifted poleward which is also associated with a small poleward displacement of the jet stream position region (Fig. 12c). In addition to small spatial shifts, the average potential temperature in the WCB outflow region differs (Fig. 12; see also Fig. 11b

and Tab. 3). The large part of WCB outflow potential temperature in WEAK and extWEAK is lower than in IDEA (Fig. 12b,d), and contrarily, potential temperature in P1.5K is larger than in IDEA (Fig. 12c), which is consistent with WCB trajectories' average maximum isentropic level. Thus, in addition to poleward/equatorward displacements also the vertical position of WCB trajectories is influenced by the modification of low-level specific humidity and temperature with WCB air parcels reaching only a lower outflow height in the experiments with weaker SST gradients and a higher outflow height in the experiment with

increased SSTs.



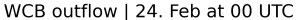

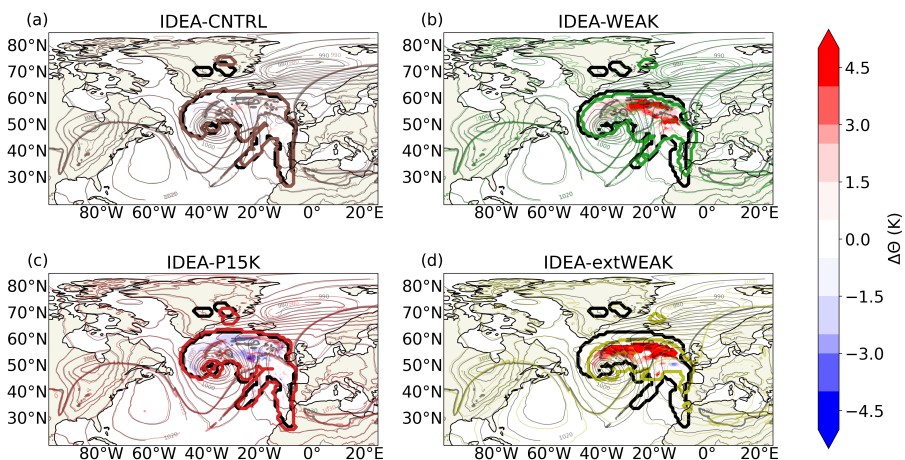

**Figure 12. (a)-(d)** Potential temperature difference ($\Delta\Theta$, shading, in K) in the WCB outflow at 00 UTC 24 February 2019 for **(a)** IDEA-CNTRL, **(b)** IDEA-WEAK, **(c)** IDEA-extWEAK, and **(d)** IDEA-P1.5K. The WCB outflow region in IDEA is outlined in black (bold contours) and the WCB outflow for the respective experiments is shown in color. Also shown are sea level pressure (contours, every 5 hPa and 2-PVU potential vorticity contour at 315 K PV for IDEA (black) and the respective experiment (color).

In the following, we illustrate how changes in WCB ascent are also reflected in the large-scale atmospheric circulation. As shown in Fig. 12, the divergent upper-tropospheric WCB outflow influences the amplification of the downstream ridge (see also, e.g., Grams et al., 2011; Pfahl et al., 2015; Steinfeld and Pfahl, 2019; Oertel et al., 2023a). Specifically, the influence of modified air-sea interactions on the large-scale flow modulated by WCB ascent is reflected in the temporal evolution of

geopotential height at 500 hPa (Z500) in the Western North Atlantic where the downstream ridge develops (40 to 70 ° N and 30 ° W to 20 ° E). After three days of lead time, the influence of WCB trajectories on the upper-level ridge becomes apparent (Fig. 13). The onset of the blocking associated with an increase in Z500 aligns with the emergence of differences in the Z500 evolution among experiments. The highest Z500 values, i.e., the strongest ridge, are present in P1.5K, while the lowest are found in WEAK and extWEAK. When the upper-level ridge is fully developed and Z500 values peak, the maximum

difference between the experiments (extWEAK and P1.5K) amounts to almost 4 gpdm (Fig. 13). Clear differences in the temporally averaged Z500 fields emerge (Fig. 14). The ridge in experiments with a weakened SST gradient (WEAK and extWEAK) is weaker than in IDEA (Fig. 14b,c), while P1.5K is characterized by overall higher Z500 values (Fig. 14d). In the context of Z500 climatological anomalies, typically ranging from 15-30 gpdm in the Euro-Atlantic sector relative to a seasonal climatology (Grams et al., 2017), the observed average differences of 4 gpdm between the P1.5K and extWEAK experiments

are noteworthy. These discernible differences underscore the sensitivity of downstream ridge development to modifications in SST in the Gulf Stream region. Our results show that changes in SST mainly impact the strength of WCB ascent, while not drastically altering the overall synoptic pattern. As such, it is unlikely to see major changes in the upper-level flow. Yet, these




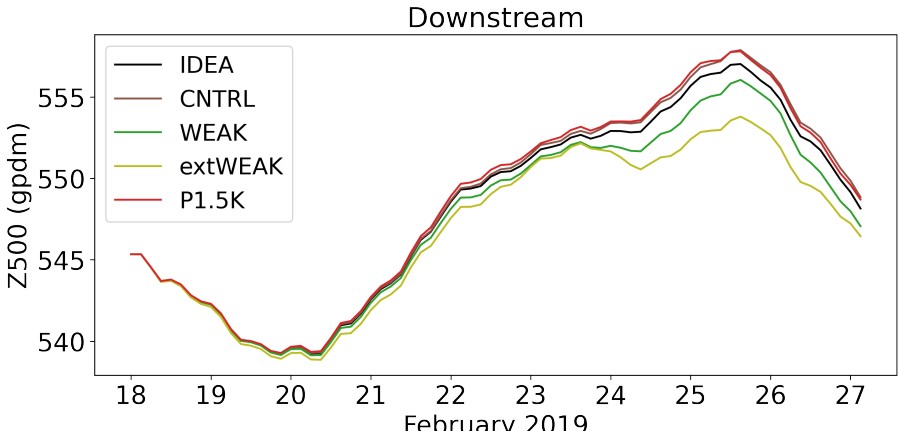

**Figure 13.** Evolution of geopotential height at 500 hPa (Z500, in gpdm) for CNTRL (brown), IDEA (black), WEAK (green), extWEAK (light green), and P1.5K (red) spatially averaged over the ridge region downstream of the Gulf Stream region (40 to 70 ° N and 30 ° W to 20 ° E).

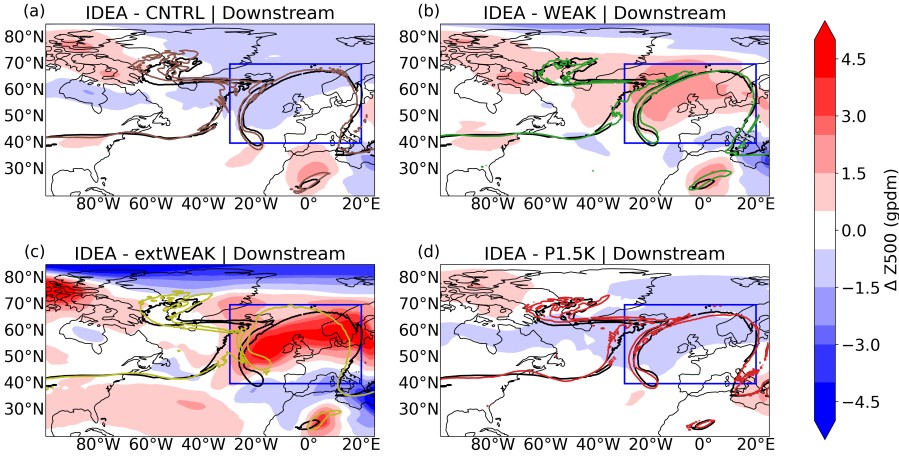

**Figure 14.** Differences in temporally averaged (21 to 27 February 2019) geopotential height (Δ Z500, shading, in gpdm) in the ridge region downstream of the Gulf Stream (40 to 70 ° N and 30 ° W to 20 ° E, green box) for **(a)** IDEA-CNTRL, **(b)** IDEA-WEAK, **(c)** IDEA-extWEAK, and **(d)** IDEA-P1.5K. 2-PVU potential vorticity contour at 315 K PV at 12 UTC 25 February 2019 for IDEA (black) and the respective experiment (in color) are shown as well.

findings suggest that blocking intensity in the Euro-Atlantic sector is sensitive to the Gulf Stream SST pattern and influenced by changes in WCB ascent characteristics resulting from modified air-sea interactions.

Overall our findings suggest that SST within the Gulf Stream region indeed can affect the downstream flow evolution by influencing the WCB inflow region (Figs. 4 and 5) through modified surface fluxes (Fig. 3), which subsequently modifies WCB



ascent characteristics (Figs. 9 and 10). These perturbations then also propagate into the mid to upper troposphere (Fig. 11), where the WCB interacts with the large-scale flow and influences the strength of the downstream ridge (Fig. 13). We conclude that the WCB airstream acts as a mechanistic link between the lower and upper troposphere and can efficiently transfer near-
surface perturbations upward, linking the air-sea interactions over the Gulf Stream with the large-scale flow.

## 5 Summary and Discussion

This study investigates the effects of SST perturbations in the Gulf Stream region on the evolution of the downstream flow during an atmospheric blocking event in February 2019. A series of sensitivity experiments with the ICON model was conducted to evaluate the impact of SST changes in the Gulf Stream region on the formation and maintenance of the downstream
block. Thereby, a particular focus is on improving our understanding of the physical processes that link near-surface processes with the upper-level flow. The sensitivity experiments include five free-running simulations: (i) a control simulation (CNTRL) with SST taken from the ECMWF IFS analysis, (ii) an experiment with an idealized SST gradient devoid of small-scale meanders (IDEA), (iii) a simulation with a reduced Gulf Stream SST gradient (WEAK), (iv) an extreme scenario for which the SST gradient in the Gulf Stream was almost completely removed (extWEAK), and (v) a simulation with increased SST while
preserving the SST gradients (P1.5K). In each of the experiments, the synoptic-scale dynamics, including cyclone formation, WCB airstream ascent, and downstream ridge development, are well represented and largely similar. However, smaller, non-negligible differences arise between experiments. These subtle changes affect both boundary layer dynamics and processes in the upper troposphere, highlighting the relevance of the representation of SST for influencing the chain of events that ultimately impact the upper-level flow evolution.

The comparison of the IDEA and CNTRL experiments, which differ only by smoothing local SST gradients, indicates that small-scale SST meanders have only a minor influence on atmospheric conditions during the nine-day simulation period. This aligns with Tsopouridis et al. (2021), who reported limited effects on cyclone dynamics after smoothing SST gradients in the Gulf Stream and Kuroshio currents. Similarly, the subtle differences between the CNTRL and IDEA experiments within this short period of only several days are consistent with Roberts et al. (2021) who suggested a minimal impact of small-scale
SST meanders on atmospheric dynamics for short lead times. Yet, the role of small ocean eddies may be more significant over longer periods. Roberts et al. (2021) indicated that while SST biases or inaccuracies may not substantially influence individual synoptic events, as demonstrated in our study, their impact could be more pronounced on sub-seasonal scales. Moreover, Roberts et al. (2022) suggested that an increased ocean resolution in atmosphere-ocean coupled simulations could lead to a more noticeable overall impact, primarily due to a better representation of the SST evolution over extended timescales. The
latter is not accounted for in our experiments as SST patterns remain constant throughout the comparatively short simulation time.

In contrast to CNTRL, larger differences compared to IDEA were found for the WEAK, extWEAK, and P1.5K experiments. In particular, differences in the strength of air-sea interactions, the characteristics of WCB ascent, and the structure of the downstream ridge are present. Differences between experiments are most apparent in the boundary layer and particularly





pronounced during the passage of cyclones, which also advect cold continental air over the relatively warm ocean surface
leading to CAOs (Fig. 15a). The low-level air which had been heated and moistened as a consequence of the passage of the
first cyclone (CY1) is subsequently drawn into the WCB of the second cyclone (CY2). The latter has also been referred to as a
'hand-over mechanism' (Papritz et al., 2021). An increase in SST in the P1.5K experiment leads to enhanced upward latent and
sensible heat fluxes, which results in larger low-level moisture. In contrast, the WEAK and extWEAK experiments with colder
SSTs south of the Gulf Stream SST front due to the reduced SST gradients are associated with reduced surface fluxes, and
thus, low-level moisture. This agrees with Small et al. (2014) who noted lower surface heat fluxes, reduced moisture content,
and colder air temperature in the lower troposphere as a response to a weakened SST gradient.

Modifications of low-level moisture are relevant for subsequent flow evolution because the availability of moisture in the
WCBs inflow region influences its ascent. WCB ascent is more pronounced and on average faster in the P1.5K experiment in
which specific humidity prior to WCB ascent is higher (Fig. 15b). During the ascent higher initial moisture availability leads to
a more efficient conversion of water vapor into hydrometeors, resulting in enhanced diabatic heating. In contrast, the WEAK
and extWEAK experiments show reduced diabatic heating as well as weaker and slower WCB ascent. This emphasizes the
importance of specific humidity content in the WCB inflow region for its ascent behavior, which agrees with previous studies
(Schemm et al., 2013; Schäfler and Harnisch, 2015; Oertel et al., 2021, 2023a; Joos et al., 2023). Moreover, differences in low-
level moisture availability, together with changes in the temperature distribution and low-level baroclinicity, influence cyclone
intensity while cyclone tracks are not affected. Correlations between moisture availability and cyclone intensity have already
been reported by Booth et al. (2012). However, it is acknowledged that the relative contributions of moisture availability, static
stability (from surface heating), or baroclinicity to cyclone intensification are not disentangled here.

SST modifications do not only influence WCB ascent behavior and trajectory number but also influence the WCB outflow
properties in the upper troposphere. Potential temperature in the WCB outflow P1.5K experiment is higher than in IDEA while
it is lower in the WEAK and extWEAK experiments, which follows from differences in diabatic heating during ascent. Thus,
WCB trajectories with positive humidity perturbations in their inflow region on average reach higher isentropic levels (see also
Oertel et al., 2023a). This impact on WCB outflow properties distinctly contrasts with the effect of stochastically perturbed
parametrization tendencies (SPPT; Buizza et al., 1999) on WCB outflow properties as reported by Pickl et al. (2023). They
found that usage of SPPT increased the number of WCB trajectories whereas the average diabatic heating or outflow properties,
such as outflow potential temperature, remained unchanged. It is noteworthy that the relatively subtle differences observed in
the upper-level ridge across the experiments can be attributed to some extent to the duration of the simulations. We hypothesize
that longer lead times may result in more pronounced discrepancies. Yet, the differences in Z500 between experiments are
still remarkable when considered relative to the range of climatological Z500 anomalies. However, we also note that for
substantially longer lead times, such as sub-seasonal to seasonal timescales, atmosphere-ocean coupled simulations or SST
updates are required.

The results presented here support previous research underscoring the link between Gulf Stream SST and downstream
blocking events in the North Atlantic and Western Europe (e.g. O'Reilly et al., 2016; Yamamoto et al., 2021). We conclude
that SST perturbations affect WCB characteristics consistently at every stage of its development, i.e. in the inflow, ascent, and




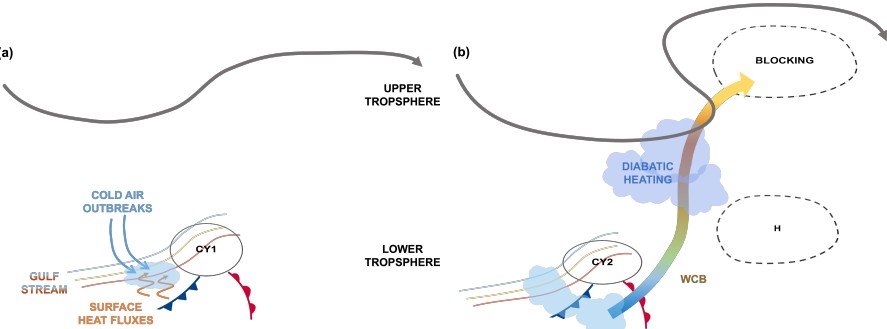

**Figure 15.** Schematic of process chain linking air-sea interactions in the Gulf Stream to the amplification of a blocking event over Europe.
**(a)** The first cyclone (CY1) triggers a cold air outbreak with associated high surface fluxes increasing low-level specific humidity (light blue shading). **(b)** The second cyclone (CY2) travels into the region preconditioned by CY1. WCB of CY2 originates in the region preconditioned by high surface fluxes and ascends leading to cloud formation and latent heating. This associated divergent WCB outflow amplifies and maintains the downstream ridge (grey dashed contour) associated with a surface high (H). The sensitivity of the blocking to SST emerges through the WCB linking humidity conditions near the surface with the upper-level flow.

outflow stages. The WCB is thus an important link that connects the SST changes in the Gulf Stream region with the dynamics in the middle and upper troposphere, thereby influencing the large-scale circulation and the strength of atmospheric blocking downstream, which is in line with Wenta et al. (2024). Fig. 15 illustrates the chain of synoptic events and various interacting processes.

While this study provides valuable insights into the underlying physical mechanisms, it is based on a specific case study
and a single model setup. Therefore, its findings cannot be universally applied to other cases and synoptic situations. This is especially relevant as the passage of several intense cyclones across the Gulf Stream region in this case increased the atmosphere's sensitivity to SST changes. A further limitation of our analysis is the usage of an atmosphere-only simulation, in which SSTs are static and not dynamically coupled with the ocean. Besides, with this setup, we were unable to determine whether the absolute SST values or the local SST gradients play a more important role for cyclone intensification and WCB
ascent. To unravel these complex processes and better understand the role of air-sea interactions, especially in coupled systems, future research is required. Furthermore, to quantify the effects of systematically perturbed lower boundary conditions in comparison to the impacts from uncertainties in initial conditions, ensemble simulations with initial condition perturbation for the experiments would be beneficial. Despite these limitations, we believe that the process-oriented approach adopted in this study significantly contributes to understanding how different SST representations can affect large-scale atmospheric flow
evolution.

*Data availability.* WCB trajectories and SST patterns from the experiments will be made publicly available in RADAR4KIT after acceptance of the manuscript.





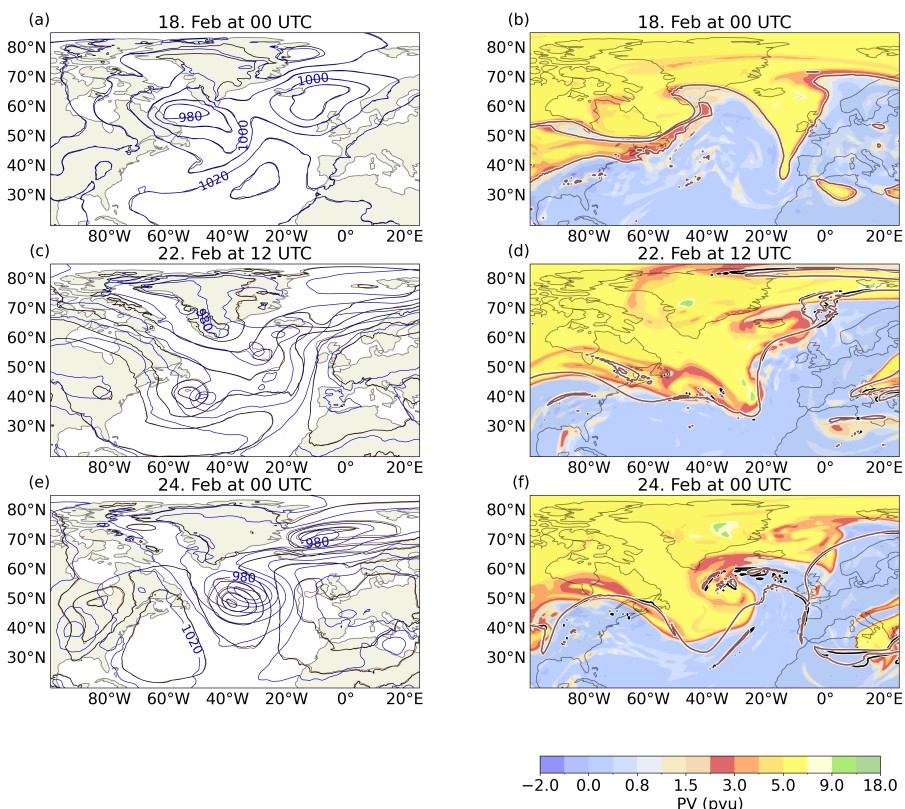

**Figure A1. (a,c,e)** Sea level pressure (contours, every 10 hPa) for ERA5 (blue), CNTRL (brown) and IDEA (black). **(b,d,f)** Potential vorticity contour (PV) at 315 K (shading in PVU) for ERA5 and 2-PVU potential vorticity contour at 315 K for CNTRL (brown) and IDEA (black). **(a,b)** 00 UTC 18 February, **(c,d)** 12 UTC 22 February, and **(e,f)** 12 UTC 24 February.

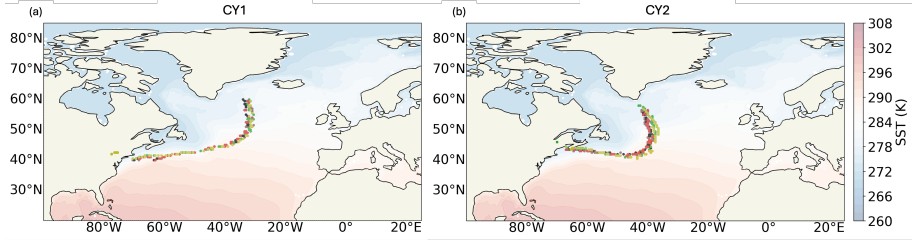

**Figure A2. (a)** Cyclone tracks of the first cyclone (CY1) from 10 UTC 18 February to 19 UTC 21 February (hourly) CNTRL (brown), IDEA (black), WEAK (green), extWEAK (light green), and P1.5K (red). Sea surface temperature (SST, shading in K, for IDEA). **(b)** same as **(a)** of the second cyclone (CY2) from 20 UTC 21 February to 09 UTC 25 February.

**Appendix**





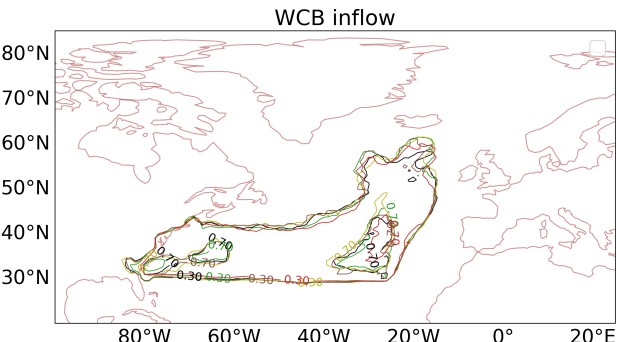

**Figure A3.** Eulerian WCB inflow region, defined as the area where the occurrence of WCB inflow (WCB trajectories' pressure larger than 800 hPa) exceeds 30 % (contours for CNTRL (brown), IDEA (black), WEAK (green), extWEAK (light green), and P1.5K (red)) during the nine-day simulation period. This area corresponds to the WCB inflow region definition used in Fig. 4 and 5. In addition, the 70 % frequency is also shown.

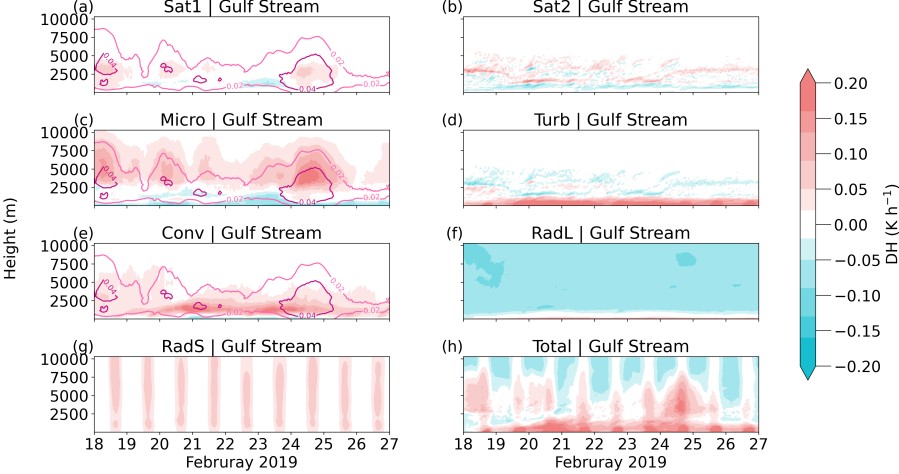

**Figure A4.** Evolution of vertical profiles of diabatic heating rates (DH, shading, in $K\,h^{-1}$) for IDEA spatially averaged over the Gulf Stream region (30 to 55 °N and 80 to 25 °W). DH rates are shown for **(a)** first saturation adjustment (Sat1), **(b)** second saturation adjustment (Sat2), **(c)** microphysics (Micro), **(d)** turbulence (Turb), **(e)** convection (Conv), **(f)** longwave radiation (RadL), **(g)** shortwave radiation (RadS), and **(h)** total DH rate. See Oertel et al. (2023a) for a detailed description and overview of heating rates. **(a, c, e)** also show total hydrometer content (pink contours, in $g\,kg^{-1}$).




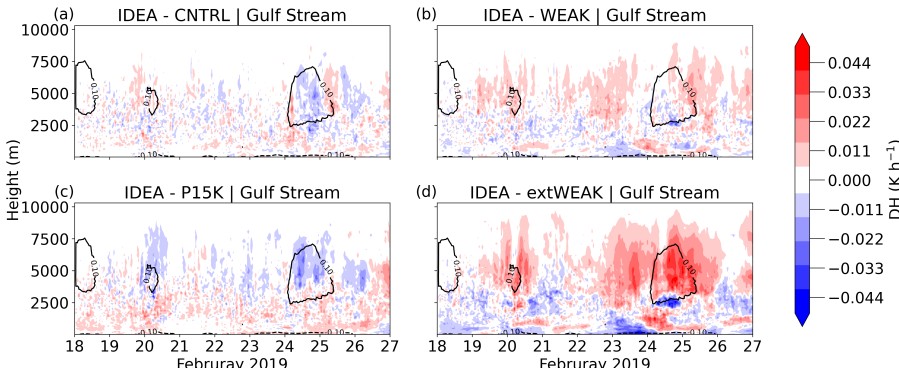

**Figure A5.** Evolution of differences of diabatic heating rate profiles related to cloud processes i.e. microphysics and first saturation adjustment (DH, shading, in $\mathrm{K\,h^{-1}}$) spatially averaged over the Gulf Stream region (30 to 55 °N and 80 to 25 °W) for **(a)** IDEA-CNTRL, **(b)** IDEA-WEAK, **(c)** IDEA-extWEAK, and **(d)** IDEA-P1.5K. Also shown is the total diabatic heating rate for IDEA (black contours, at 0.1 $\mathrm{K\,h^{-1}}$, for IDEA).

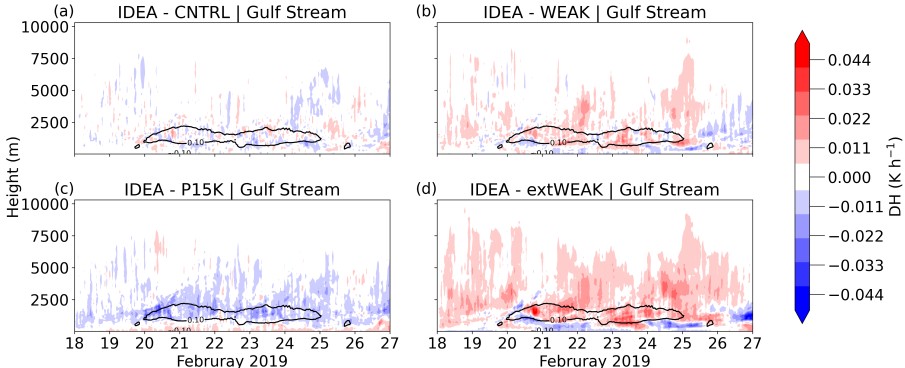

**Figure A6.** As Fig. A5 but for diabatic heating rates from convection scheme (DH, shading, in $\mathrm{K\,h^{-1}}$) spatially averaged over the Gulf Stream region (30 to 55 °N and 80 to 25 °W) for **(a)** IDEA-CNTRL, **(b)** IDEA-WEAK, **(c)** IDEA-extWEAK, and **(d)** IDEA-P1.5K. Also shown is the convective heating rate for IDEA (black contours, at 0.1 $\mathrm{K\,h^{-1}}$).



*Author contributions.* SC, AO, and MW performed the analyses, simulations, and trajectory calculations. SC wrote the first draft of the
manuscript. All authors contributed to the discussions and revisions of the manuscript.

*Competing interests.* At least one of the (co-)authors is a member of the co-editors of Weather and Climate Dynamics. The authors have no
other competing interests to declare.

*Acknowledgements.* The research leading to these results has been done within the subproject B8 of the Transregional Collaborative Research
Center SFB / TRR 165 "Waves to Weather" (www.wavestoweather.de) funded by the German Research Foundation (DFG). Moreover,
MW is supported by DFG (grant no. GR 5540/2-1). CMG was supported by the Helmholtz Association as part of the Young Investigator
Group "Sub-seasonal Predictability: Understanding the Role of Diabatic Outflow" (SPREADOUT, grant VH-NG-1243). The contribution
of AO was partly carried out within the Italia – Deutschland science-4-services network in weather and climate (IDEA-S4S; INVACODA,
4823IDEAP6). This Italian-German research network of universities, research institutes and Deutscher Wetterdienst is funded by the BMDV
(Federal Ministry of Digital and Transport). The authors acknowledge support by the state of Baden-Württemberg through bwHPC. The
ICON simulations were carried out on the supercomputer HoreKa at Karlsruhe Institute of Technology, Karlsruhe, which is funded by the
Ministry of Science, Research and the Arts Baden-Württemberg, Germany, and the German Federal Ministry of Education and Research. We
are very grateful to Nedjeljka Žagar for the discussions and her feedback on this study.



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
