# Peer review of "From Sea to Sky: Understanding the sea surface temperature impact on an atmospheric blocking event using sensitivity experiments with the ICOsahedral Nonhydrostatic (ICON) model."

_EGUsphere, 2024_

## Author Response (AR1)

**Response to reviewer comments for**

From Sea to Sky: Understanding the sea surface temperature impact on an atmospheric blocking event using sensitivity experiments with the ICOsahedral Nonhydrostatic (ICON) model
by Svenja Christ, Marta Wenta, Christian M. Grams, and Annika Oertel

We would like to express our sincere gratitude to both anonymous reviewers for their positive, detailed, and constructive feedback. Their insightful comments have been invaluable in improving the quality of this manuscript, and we appreciate the time and effort they invested in critically assessing our work. Based on their suggestions, we have carefully revised the manuscript and addressed each comment. The following document outlines our responses to the specific comments and highlights the key changes in the updated version of the manuscript. Our replies to reviewers' comments are included in blue below. Line numbers refer to the track-change version of the manuscript.

**1st reviewer response**

Review of "From Sea to Sky: Understanding the sea surface temperature impact on an atmospheric blocking event with the ICON model"

This study uses idealized model sensitivity experiments to investigate the impact of Gulf Stream sea surface temperatures (SSTs) on the development of an intense atmospheric blocking event over Europe that occured during February 2019. The authors combine traditional eulerian diagnostics with analysis of lagrangian trajectories to diagnose the response of the upper-level blocked circulation to rapidly ascending air streams in extratropical cyclones (i.e. Warm Conveyor Belts) and their sensitivity to changing SST boundary conditions. The authors provide a compelling argument that, at least for this case study, Gulf Stream SSTs and associated gradients can substantially modify the development of the upper-atmosphere circulation anomalies through their impact on air-sea fluxes and the temperature and moisture of the WCB inflow regions. The manucript is clearly written and will be of interest many readers of Weather and Climate Dynamics. I have several comments that should be straightforward for the authors to address, but otherwise I think this manuscript is suitable for publication in WCD.

Dear reviewer, thank you very much for your insightful and comprehensive feedback. A key point raised by the reviewer concerns the use of prescribed SST boundary conditions, as opposed to a coupled ocean-atmosphere model or time-varying SST boundary conditions for a forecast range of nine days. We acknowledge the significance of this issue and have ensured that our discussion of this limitation is appropriately addressed in the revised manuscript.

**Main Comments**

(1) One limitation of this study is the use of single-member deterministic sensitivity experiments for a single start date (00 UTC 18 February 2019) and the lack of uncertainty estimates. Ensemble forecasts (or analysis of multiple case studies/start dates) would provide additional uncertainty information that would allow the authors to better distinguish between systematic effects and chaotic variations that are consequence of the intrinisic predictability limits. However, I appreciate that not all research groups have access to significant HPC resources required to run such experiments. The authors should acknowledge these limitations and also emphasise those aspects of their results that give confidence that the diagnosed impacts are systematic. For example, the consistent differences between experiments at all lead times (e.g. figure 9b and others) and the physically plasuible (thermo)dynamical interpretation give me some confidence that the impacts in this cases study represent signal rather than noise. However, I think the authors could acknowledge these limitations and justify their approach more explicitly in the **intro/discussion/conclusions.**

Thank you for raising this limitation and your thoughts on this aspect. We revised the text and more explicitly discussed this limitation and justified the robustness of our results. Indeed, it would be very interesting to run ensembles for each prescribed SST boundary conditions, however, computationally resources for the simulations as well as for the subsequent analysis are limited.
As the reviewer mentioned in the reply, the consistency of the difference between simulations in line with the (thermo)dynamical processes provide confidence that the impacts are physical and not just noise. Nevertheless, we address these limitations in the revised manuscript (see lines 117, 399ff, 575ff).

(2) Another potential limitation, acknowledged by the authors, is the potential role for ocean coupling. I think it is justifiable to use a prescribed SST boundary condition to explore the sensitivity to different SST anomalies. However, I am not sure why the authors use a constant (i.e. persisted) SST throughout the 10-day forecasts rather than a time-varying boundary condition. I would like the authors to comment on whether they expect either time-varying SST boundary condition or coupled ocean feedbacks might amplify or damp the diagnosed impacts? For instance, how would ocean feedbacks impact the estimated surface turbulent heat fluxes that are important for modifying the properties of the marine atmospheric boundary layer than feed WCBs? Perhaps one possibility is to compare the fluxes from ICON sensitivity experiments with those from a coupled NWP forecast of the same case (e.g. from operational ECMWF forecasts, which have been coupled in ENS/HRES since 2018).

We agree with the reviewer's concerns. A limitation of this analysis is the prescribed SST boundary conditions throughout the 9-day simulation, which has been mentioned in the manuscript (line 562f: "A further limitation of our analysis is the usage of an atmosphere-only simulation, in which SSTs are static and not dynamically coupled with the ocean.")
As our key question addresses the sensitivity to different SST anomalies, we decided to use fixed prescribed SST conditions to control SST and the SST gradient throughout the simulation. We

acknowledge that state-of-the-art forecasts for such lead times currently employ a coupled ocean model, however, our idealized SST conditions are not realistic but target specifically the sensitivity to the underlying SST conditions as well as the propagation of these perturbations through the atmosphere to the upper-level jet level. We believe that the key conclusions of the manuscript and the identified physical mechanisms remain valid regardless of the interactive ocean. Yet, for future analysis, it would be interesting to include a coupled ocean with subsequently artificially smoothed SST gradient. We have included these aspects in the discussion in the revised manuscript.

To estimate the potential amplifying or damping mechanism of an interactive ocean, we have compared the evolution of SST and the surface heat fluxes in the Gulf Stream region between our CNTRL simulation/experiments and the ERA5 IFS daily varying SST (Hersbach et al., 2020; and see Fig. R1 below in this replay document).  As shown in the manuscript in Section 3, the synoptic evolution between CNTRL and ERA5 is very similar, which allows for a comparison of *realistic* ERA5 latent and sensible heat fluxes in this case study and the ICON simulations (see also Fig. R2).

The spatial mean across the Gulf Stream region shows the same temporal evolution, which is mainly determined by the passage of the cyclones as discussed in the manuscript (Section 4.1). Moreover, the variance of SST is of the same order of magnitude as in the constant SSTs in our experiments (see also Figure R1). Thus, there is no evidence for substantially larger or smaller heat fluxes resulting from the interaction of the atmosphere and the ocean in the re-analysis, at least averaged across the North Atlantic region. Yet, on the local scale, cyclone passages can influence the ocean, e.g. by cooling SST (Dacre et al., 2020), which would reduce the surface fluxes and is somewhat consistent with the small difference between ERA5 and CNTRL (Fig. R1).

Therefore, we argue that for the purpose of our senstivity experiments it is justifiable to use constant SSTs. Even though we agree with the reviewer's concern and acknowledge that in this study we miss the effect of air-sea interactions on the ocean circulation and cooling of the SST after a cyclone passed over, although this might be less relevant in the warm sector of the cyclone.

We have extended the discussion of this limitation in the revised manuscript (lines 563ff).

[Figure]

Fig. R1 Sea surface temperature (SST, in K) bold lines spatially averaged over the Gulf Stream region (30 to 55°N and 80 to 25°W) and dotted lines standard deviation from the mean for ERA5 (blue, daily SST analysis from OSTIA) and CNTRL (brown). (b) as (a) but for latent heat flux (LHF, in Wm$^{-2}$, upward defined negatively). (c) as (a) but for sensible heat flux (SHF, in Wm$^{-2}$).

[Figure]

Fig. R2 Sea surface temperature (SST, in K) spatially averaged over the Gulf Stream region (30 to 55°N and 80 to 25°W) for ERA5 (blue, daily SST analysis from OSTIA), CNTRL (brown), IDEA (black), WEAK (green), extWEAK (light) green, and P1.5K (red). (b) as (a) but for latent heat flux (LHF, in Wm$^{-2}$, upward defined negatively). (c) as (a) but for sensible heat flux (SHF, in Wm$^{-2}$).

(3) It is not clear to me why "IDEA" is chosen as a reference experiment for comparisons with other sensitivity experiments rather than "CNTRL". I think the comparison of "IDEA" and "CNTRL" is scientifically interesting as it gives some indication of the sensitivity of the atmospheric circulation to the presence of mesoscale ocean eddies. The relative magnitude of the difference between "IDEA" and "CNTRL" is also useful context for the other comparisons, which could be a useful benchmark when considering the significance of results as described in (1).

As the reviewer pointed out, the choice of the reference experiment was not clear. First of all, the presented results would not change much if CNTRL (instead of IDEA) were chosen as reference experiment because both IDEA and CNTRL simulations differ only very slightly (see Section 4 of the manuscript). The idea behind using IDEA as reference experiment was to have a benchmark with a semi-idealized SST gradient (IDEA) and prescribed controlled differences to this set, i.e., WEAK, extWEAK, and P1.5K are derived from IDEA, while CNTRL includes the original unmodified SST data. For example, the difference between P1.5K and IDEA is only the constant SST difference of 1.5 K, while compared to CNTRL it is a temperature offset plus the removal of small-scale SST features.
To address the atmospheric sensitivity to the presence of mesoscale ocean eddies, we extended the discussion on the differences between IDEA and CNTRL (see lines 256ff, 505ff, 516f).

**Minor comments**

Figure 1 - the differet SST gradient contours are difficult to distinguish in panel (a).

Thanks, for your comment, we tried to improve the SST contours in panel (a)

Table 1 - Please state the original source of the SST used in the IFS analysis. SST is not part of the IFS 4DVar so it is taken from another high-resolution satellite product (I think OSTIA).

Thanks for this helpful comment, as you suggested the SST is taken from the high-resolution satellite product OSTIA. We have included this information in the manuscript (line 150ff).

Figure 2 - It is very difficult to distinguish trajectories coloured by pressure and PV contours. Perhap separate into separate panels for PV and trajectories?

We agree.  As we like to show the trajectories in the same figure as the synoptic evolution, we changed the colormaps to improve visibility (see also comment by reviewer 2).

Line 272 - "a decrease in surface heat fluxes" -> "reduced magnitude (i.e. less negative) heat fluxes"?

Thanks for the suggestion, we changed the expression from "a decreases in surface heat fluxes" to "reduced magnitude (i.e. less negative) heat fluxes"

Line 292 - Is the WCB inflow region fixed across all lead times or time varying?

Thanks for this helpful question. The WCB inflow region is fixed across all lead times. A time-invariant inflow region is justifiable because the WCB inflow from a Lagrangian perspective occurs repeatedly in this region. We clarified the description in the manuscript (lines 296ff).

Figure 4 - I suggest removing "of vertical profiles" from the caption to read "Evolution of air temperature differences [...] spatially averaged over the fixed Eulerian WCB inflow region". I initially interpreted this as the averages along the lagrangian trajectories.

Thank you for the suggestion, we changed the caption accordingly.

Line 380-382 - The authors describe the difference in WCB trajectory numbers between IDEA and CNTRL as "relatively small" but it is non-zero and the discussion could be more quantitative. From Table 3 it seems that removing the small scale eddies reduces WCBs by 4%, which is not negligible compared to the 9% reduction with WEAK. The authors could comment more about the potential impact of small-scale ocean eddies on the upper-level circulation, which has been suggested in other publications. e.g. "Ocean fronts and eddies force atmospheric rivers and heavy precipitation in western North America" - https://www.nature.com/articles/s41467-021-21504-w.

Thanks for this comment. We agree, and more thoroughly discuss the differences between IDEA and CNTRL and tried discuss more quantitative and to comment on the potential impact of small-scale ocean eddies (lines 278, 286ff,  329f, 349f, 393ff, 407f, 415f, 429f, 466ff, 477, 504, 505ff). The mentioned publication has also been included.

**2nd reviewer response**

"From Sea to Sky: Understanding the sea surface temperature impact on an atmospheric blocking event using sensitivity experiments with the ICOsahedral Nonhydrostatic (ICON) model"

By S. Christ et al.

Submitted to EGUsphere

The author presents work on using the ICON model for sensitivity experiments on the impact of SST fronts, namely how the smoothing of these fronts (as well as an increase in SSTs) will impact atmospheric blocking events, extratropical cyclone development (downstream and upstream), as well as general air-sea interactions. The research presented here is extremely well thought out, dense, well executed, and with significant literature for their background, as well as comparing their results to previously published papers. I believe the science is quite sound, the results are fascinating, and it is worthy of acceptance and publication.

My recommendations are mainly technical and with the figures; I believe they will help future readers better understand the work these authors present.

Dear reviewer, thank you very much for your positive and encouraging feedback on our manuscript. We greatly value your thoughtful recommendations, particularly regarding the technical aspects and figures input and agree that enhancing these elements will help improve clarity for future readers. We carefully implement your suggestions.

Figure 1: Flip Figures 1d (P1.5K) and 1e (extWEAK). In the rest of the paper, P1.5K is referred to after the WEAK and extWEAK cases. In order to be consistent, I think it is best to list them in the same order in this figure.

Thanks, for noting this small inconsistency. We changed the order in Figures 1 to be consistent Throughout the manuscript.

Line 172-177 (P1.5K description): Is this case study adding the 1.5K SST increase to the IDEA or CNTRL case? Or a mixture of the two? It seems similar to IDEA, but it wasn't clear which experiment this temperature increase was being applied to, or if this should be treated completely separately from the rest.

We have clarified the descriptions of P1.5K (line 177). The temperature increases was applied to the IDEA experiment, which is also one reason why we use IDEA as a reference (see also reply to comment (3) by reviewer 1).

Figure 2:

- Both on screen and printed out, it is difficult to see the trajectories in Figure 2c,f,i as some of the colors for the trajectories and background PV overlap or their hues are close. I do agree it's important to have these in the same figure, but I would suggest colormaps that do not overlap so details are not missed.
- Add large headers at the top of each column/left of each row (i.e., SST for Column 1, CY1 for Row 1, and so on) and only keep the date/time above each individual figure.
- Double check the dates and times listed in the figure description (a,c,e), as this does not appear to line up with the panels.

Following the reviewer's suggestion, we adjusted the dates and times, changed the colormaps and included headers in the figures.

Figure 3, 6, 7, 9, 10, 11, 13: The light green line for extWEAK can sometimes be difficult to see (especially in the blue shading to indicate CY1 and CY2). I would suggest a different color or line type (i.e. dashed/dotted) so that it is easier to view.

Thanks for your suggestions, we agree and tried to improve the visibility of the extWEAK experiment.

Figures 4 and 5: While the light blue shading for CY1 and CY2 works in other figures, it clashes with the colormap here. I would suggest a different color or hash marks to indicate when CY1/2 occurs.

Thanks for your comment, we changed the color.

Figure 8:

- The letters indicating each figure are hard to see. I would increase the size and reposition them
- The description for Figures 8c and 8d appears to be flipped
- The shadings in Figure 8c are too similar to each other. I would pick a different color for liquid or ice.

We agree with the reviewer and improved the outline/color schemes of Figure 8.

Figures 10 and 11: The overlapping shading for the mean standard deviation on every single line is distracting and confusing. Remove the shading and have only lines indicating the ±σ.

Thanks for this helpful comment, we adjusted the figure accordingly.

**References**

Dacre, H. F., Josey, S. A., and Grant, A. L. M.: Extratropical-cyclone-induced sea surface temperature anomalies in the 2013–2014 winter, Weather and Climate Dynamics, 1, 27–44, https://doi.org/10.5194/wcd-1-27-2020, 2020.

Hersbach, H., Bell, B., Berrisford, P., Hirahara, S., Horányi, A., Muñoz-Sabater, J., Nicolas, J., Peubey, C., Radu, R., Schepers, D., Simmons, A., Soci, C., Abdalla, S., Abellan, X., Balsamo, G., Bechtold, P., Biavati, G., Bidlot, J., Bonavita, M., De Chiara, G., Dahlgren, P., Dee, D., Diamantakis, M., Dragani, R., Flemming, J., Forbes, R., Fuentes, M., Geer, A., Haimberger, L., Healy, S., Hogan, R. J., Hólm, E., Janisková, M., Keeley, S., Laloyaux, P., Lopez, P., Lupu, C., Radnoti, G., de Rosnay, P., Rozum, I., Vamborg, F., Villaume, S., and Thépaut, J.-N.: The ERA5 global reanalysis, Q. J. Roy. Meteorol. Soc., 146, 1999–2049, https://doi.org/10.1002/qj.3803, 2020.